# Characterizing Uncertainty in the Hydraulic Parameters of Oil Sands Mine Reclamation Covers and its Influence on Water Balance Predictions

M. Shahabul Alam[1], S. Lee Barbour[1,2], Mingbin Huang[1,3]

[1]Department of Civil, Geological and Environmental Engineering, University of Saskatchewan, Saskatoon, SK, S7N 5A9, Canada
[2]Global Institute for Water Security, University of Saskatchewan, Saskatoon, SK, S7N 3H5, Canada
[3]Center for Excellence in Quaternary Science and Global Change, Chinese Academy of Sciences, Xian 710061, China

*Correspondence to*: M. Shahabul Alam (msa181@usask.ca)

**Abstract.** One technique to evaluate the performance of oil sands reclamation covers is through the simulation of long-term water balance using calibrated soil-vegetation-atmosphere-transfer models. Conventional practice has been to derive a single set of optimized hydraulic parameters through inverse modelling (IM) based on short-term (<5-10 y) monitoring datasets. This approach is unable to characterize the impact of variability in the cover properties. This study utilizes IM to optimize the hydraulic properties for 12 soil cover designs, replicated in triplicate, at Syncrude's Aurora North mine site. The hydraulic parameters for three soil types (peat coversoil, coarse-textured subsoil, and lean oil sand substrate) were optimized at each monitoring site from 2013-2016. The resulting 155 optimized parameter values were used to define distributions for each parameter/soil type, while the Progressive Latin Hypercube Sampling (PLHS) method was used to sample parameter values randomly from the optimized parameter distributions. Water balance models with the sampled parameter sets were used to evaluate variations in the maximum sustainable leaf area index (LAI) for five illustrative covers and quantify uncertainty associated with long-term water balance components and LAI values. Overall, the PLHS method was able to capture broader variability in the water balance components than a discrete interval sampling method. The results also highlight that climate variability dominates the simulated variability in actual evapotranspiration, and that climate and parameter uncertainty have a similar influence on the variability in net percolation.

## 1 Introduction

The hydraulic parameters of reclamation soil covers on oil sands mine waste have most commonly been characterized by calibrating water dynamics models against a single profile of field-monitored water content and suction. In many cases, this has been undertaken by deriving a single set of optimized parameter values from inverse modelling (IM) of short-term (5-10 years) monitoring data (Alam et al., 2017; Boese, 2003; Huang et al., 2015, 2011a, b, c; Keshta et al., 2009; Price et al., 2010; Qualizza et al., 2004). Devito et al. (2012) recommend that model calibration be focused on seasonal and inter-annual climate

variability (e.g., wet or dry) and also take into account spatial variations in water movement within a spatially heterogeneous landscape. The current modelling approach that attempts to determine a single set of 'best fit' properties based on IM of a single monitoring station is unable to characterize the spatial or temporal variability within the hydraulic properties of the cover soil and underlying mine waste. Quantifying spatial and temporal variability would be of value when assessing the expected long-term performance of reclaimed oil sands closure landscapes. However, spatial and temporal variability (i.e., uncertainty) in model parameters are not conventionally quantified or incorporated in the soil-vegetation-atmosphere-transfer (SVAT) models used to simulate long-term cover performance.

The focus of this study is characterization of the uncertainty in the hydraulic parameters of reclamation soil covers over oil sands mining waste and the impact of this uncertainty on predictions of the long-term water balance for these sites. The two key measures of success for oil sands mine reclamation are the water balance components of actual evapotranspiration (AET) and net percolation (NP). AET quantifies the ability of the cover to support re-vegetation, while NP quantifies recharge into the underlying mine waste and the concomitant impact on water and contaminant release to downgradient surface water bodies. The temporal variability of hydraulic parameters for these cover soils has been characterized by both direct testing and IM. Temporal variability in hydraulic conductivity (Ks) was measured in reclamation covers over saline-sodic overburden at Syncrude's Mildred Lake mine by Meiers et al. (2011), and a similar evolution in Ks was also obtained through IM by Huang et al. (2015). Such observed temporal variability was assumed to be the result of changes in density and pore-size distribution of reclamation soils as a result of freeze/thaw or wet/dry cycles and vegetation establishment. Spatial variability would be expected to occur in reclamation covers because of variations in soil texture, cover construction/placement conditions, topography, or vegetation establishment. For example, Huang et al. (2016) characterized the spatial variability of Ks using air-permeability testing of covers.

More recently, IM modelling has been undertaken on multiple monitoring sets collected over multiple years to evaluate the impact of parameter variability on the predicted long-term performance of reclamation covers (Alam et al., 2017; Huang et al., 2017; OKC, 2017). Recently, Alam et al. (2017, 2018b) undertook a preliminary evaluation of the impact of variability in the hydraulic properties of reclamation covers on the long-term water balance of oil sands reclamation covers. In that study, IM modelling using HYDRUS-1D was undertaken for four different reclamation covers (replicated in triplicate) over three monitoring years to characterize the water retention and hydraulic conductivity of the covers. The calibrated (optimized) parameters showed that parameter variability could be linked to both spatial and temporal variability but was dominated by spatial variability. A key limitation of this previous study was that the variability in the hydraulic properties was represented only by discrete values (i.e., the mean value of the parameter as well as upper and lower bounding values) without a full statistically based characterization of the parameter variability. The value of a full statistical description of variability in characterizing the uncertainty in the predicted water balance of the covers under a prescribed, future, climate variability was unknown.

The use of soil hydraulic parameters with spatial and/or temporal variability instead of a single parameter set can provide more information about prediction uncertainty associated with watershed response to climate variability (Benke et al., 2008). Various Monte Carlo (MC)-based approaches [e.g., generalized likelihood uncertainty estimation (GLUE; Beven and Binley, 1992), Metropolis algorithm, and Monte Carlo Markov Chain (MCMC; Metropolis et al., 1953)] can be used to sample parameters

randomly from the posterior distributions of the optimized parameters. Given that MC-based sampling strategies can be computationally expensive and sometimes unaffordable for computationally demanding models, other sampling strategies have been developed and improved over the last several decades. Of these, Latin Hypercube Sampling (LHS; McKay et al., 1979) has been most commonly used for uncertainty and sensitivity analysis in the field of water and environmental modelling (Hossain et al., 2006; Gong et al., 2015; Higdon et al., 2013; Sheikholeslami and Razavi, 2017). The LHS approach offers a

sampling strategy that can significantly reduce the sample size without compromising the accuracy of uncertainty estimation compared to the MC sampling approach (Iman et al., 1980, 1988; McKay et al., 1979). However, a major drawback of traditional LHS- and MC-based sampling strategies is that the entire sample set is generated together and, unfortunately, an appropriate sample size is not known a priori. An appropriate sample size here refers to a sufficiently large number of sampled parameters so as to achieve convergence towards a common mean and standard deviation (SD) of the parameters as well as

the mean and SD of major components of the water balance (e.g., AET and NP).

The appropriate sample size for each parameter can be determined using a convergence criterion in the case of LHS; however, the whole sample size is discarded if the convergence criteria fail, and a new set of simulations must be conducted with a larger sample size to achieve convergence. To overcome this computational demanding approach, a new, efficient, and sequential sampling strategy called Progressive Latin Hypercube Sampling (PLHS; Sheikholeslami and Razavi, 2017) can be used. In

PLHS, the sample size is divided into a series of smaller subsets (in place of the single sample set used for LHS), and each subset is added progressively to sequentially grow the sample size. This can be summarized as follows: (i) each smaller subset forms a Latin hypercube, (ii) the progressively added subsets form a Latin hypercube, and (iii) the entire sampled parameter set (consists of all smaller subsets) also forms a Latin hypercube. The details on LHS and PLHS are provided in Appendix A. The two key advantages of the PLHS method over other sampling methods (e.g., MC) are as follows: (i) it achieves a given

convergence criterion with a smaller number of samples (i.e., smaller sample size) and (ii) it allows for sequential sampling without having to discard the whole sample size when convergence criteria are not attained.

The key research question of this study is as follows: What is the influence of soil hydraulic parameter uncertainty on the long-term cover performance of the reclamation covers in northern Alberta, Canada. This question led us to the following study objectives: (i)Identify the most-efficient way to characterize distributions of the optimized hydraulic parameters from a

physically-based water balance model for an oil sands reclamation covers in northern Alberta, Canada and (ii) Quantify relative uncertainty from various sources associated with the long-term water balance of the reclamation covers.

These objectives will be met by undertaking IM of multiple monitoring sets (multiple monitoring sites in multiple years) to develop a statistical distribution of parameter variability. These distributions will be primarily utilized within a PLHS-based

sampling approach to predict variability in the expected performance of the covers over the long term based on SVAT modelling. Comparisons will determine how these predictions differ if either a discrete or continuous distribution function is used to characterize material variability. To the best of our knowledge, this more rigorous approach to evaluating the long-term performance of soil covers has not been conducted in general or specifically applied to the evaluation of oil sands reclamation covers.

## 2 Materials and methods

### 2.1 Study sites and reclamation covers

This study used soil monitoring data and meteorological data collected from the Aurora Soil Capping Study (ASCS), located at the Aurora North Mine of Syncrude Canada Ltd. (SCL) in Alberta, Canada (Fig. 1a). The ASCS is comprised of a series of 12 alternate, 1 ha covers, replicated in triplicate, and placed over a lean oil sands (LOS) overburden dump. The primary purpose of the different cover designs was to compare the performance of alternate materials and cover thicknesses in supporting vegetation and net percolation (OKC, 2017). The layout of the 12 covers (replicated) are shown in Fig. 1b and are designated by a treatment number (i.e., TRT #), with each treatment having three replicate cells for a total of 36 cells in the ASCS that were randomly placed across the watershed.

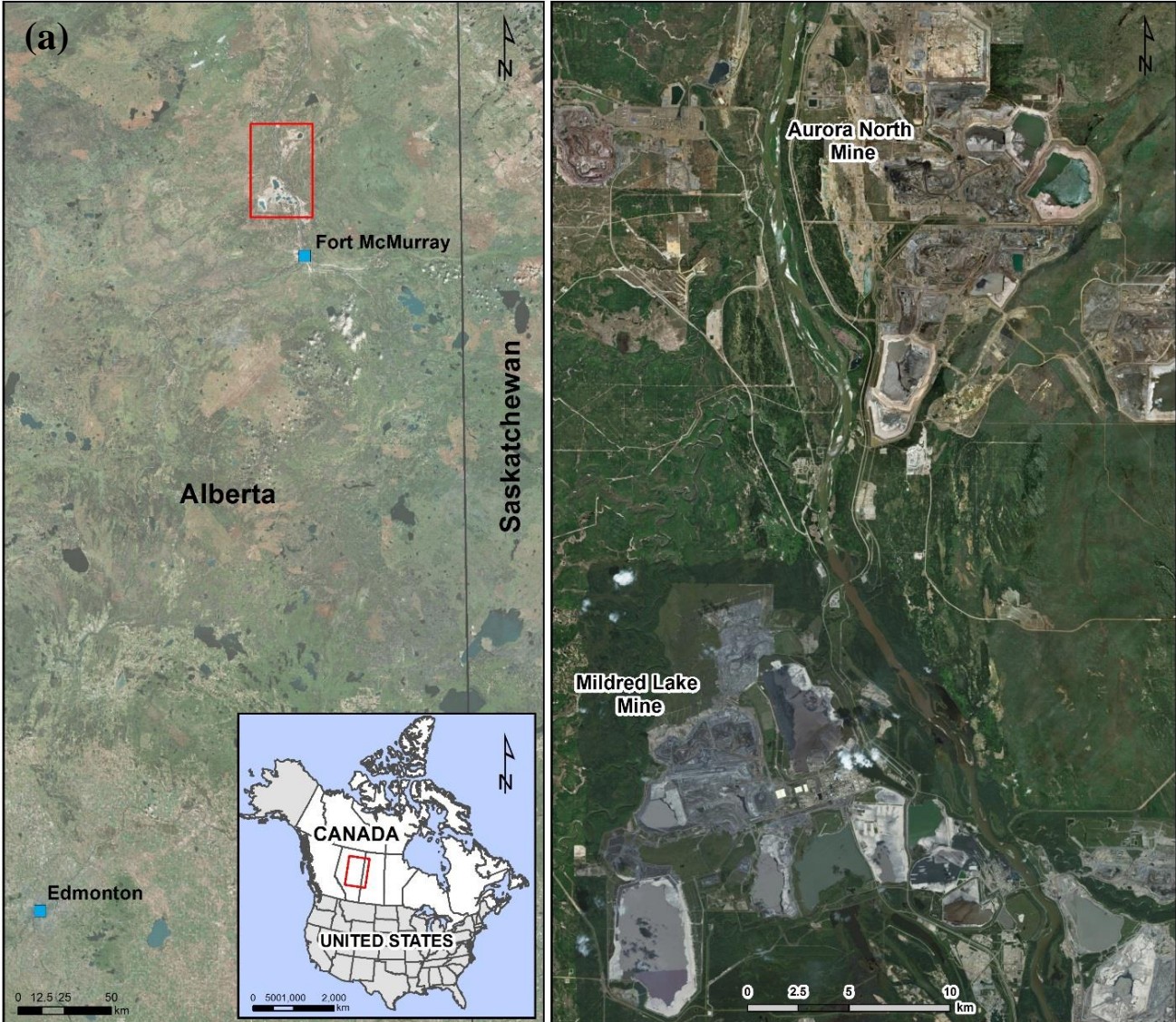

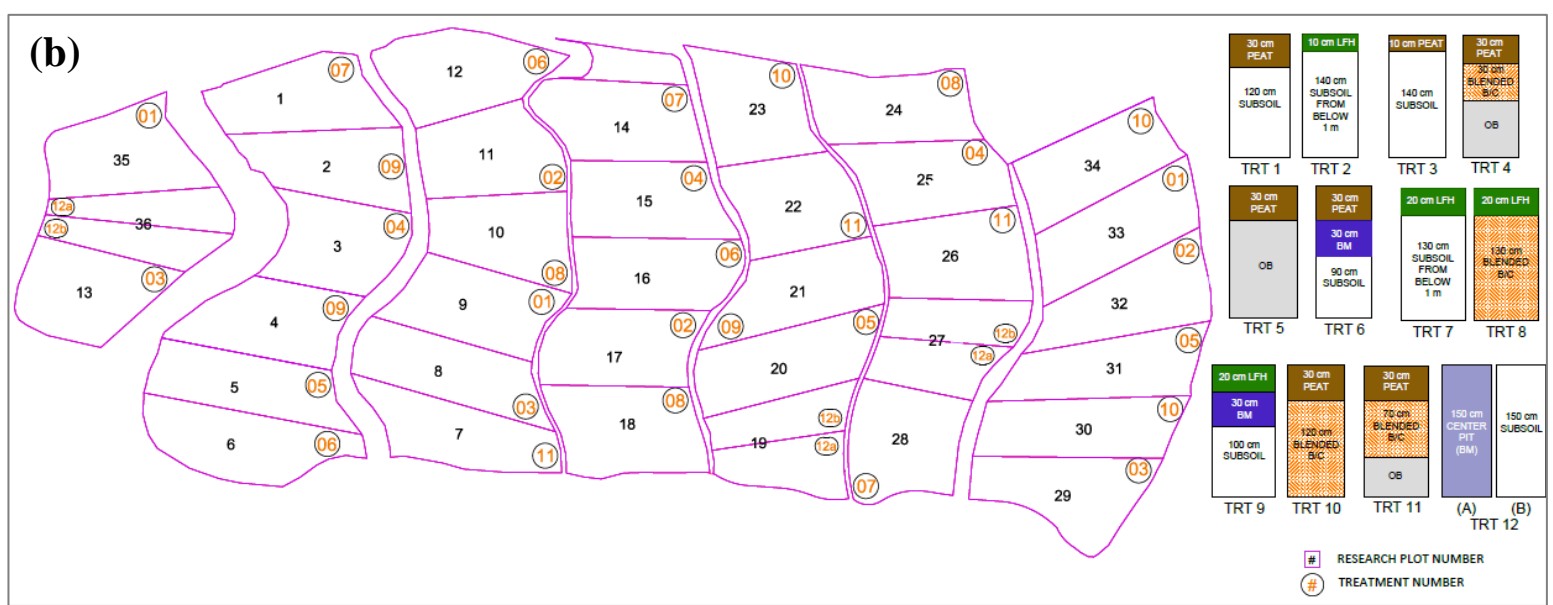

Figure 1: (a) Location map of Aurora North Mine of Syncrude Canada Ltd. and (b) soil cover design treatments (TRT) at ASCS (adapted from OKC, 2017). LOS overburden (OB) underlies all treatments, even though treatments with less than 150 cm total soil cap thickness only show OB

All the treatment covers within the ASCS were constructed in 2012 using three distinct soil layers including: coversoil, subsoil, and LOS. The coversoil was utilized in the treatment covers was either salvaged peat or LFH material. The Soil Classification Working Group in Canada defined LFH as "organic soil horizons (L, F, H) developed primarily from the accumulation of leaves, twigs and woody materials, with or without a minor component of mosses, that are normally associated with upland forest soils with imperfect drainage or drier". The L, F, and H horizons are characterized by the accumulation of original organic matter, partially decomposed organic matter, and decomposed organic matter, respectively. The peat was predominantly organic material with a total organic carbon of about 17% (by weight), while the general texture of the mineral component of LFH was sand (about 92% by mass). The coversoil was underlain by different selected coarse-textured subsoils salvaged from different locations (i.e. depositional environments) and depths within the mine site (Soil Classification Working Group, 1998). In general, the subsoil texture is sand (92%-95% by mass). The bottom layer was constructed using LOS overburden materials that was overlain by coversoil and subsoil layers. The LOS materials consist of loamy sand to sandy loam with an oil content of 0.1% to 7.7% (NorthWind Land Resources Inc., 2013). Overall, the LOS comprises a range of different oil contents and particle sizes compared to the coversoil and subsoil materials. All of the 13 treatment covers (which include two sub-categories of TRT 12) were included in this study.

Particle size distribution (PSD) analyses of the cover soil [LFH and peat], subsoil, and LOS were performed by a commercial laboratory ( OKC, 2009) in November of 2009 based on ASTM standard testing method D422 (ASTM, 1998). The ASTM D422 method is based on the assumption that the particles are spherical in shape, so the PSD for peat may not be representative. The PSD for the LFH and peat coversoils, coarse-textured subsoil, and LOS are presented in Fig. 2. The PSDs for the subsoils are the most variable, being salvaged from different depths and depositional environments located on the Aurora North mine site. For the purposes of the IM, the peat and LFH coversoils were ultimately combined into a single group as were different salvaged subsoils and LOS overburden materials. Combining the soil layers in this manner produces additional variability within each grouping; however, it ensures the maximum number of observations are utilized to capture the variability associated with each layer of the soil reclamation covers. According to Syncrude Canada Ltd., in the final cover design the top layer might be either peat/LFH or combination of the two, the distributions of parameters for these two materials together seem reasonable to be used in the illustrative covers for long-term simulation of water balance components. Therefore, the PLHS method was used to randomly sample from the distributions of the two materials grouped together.

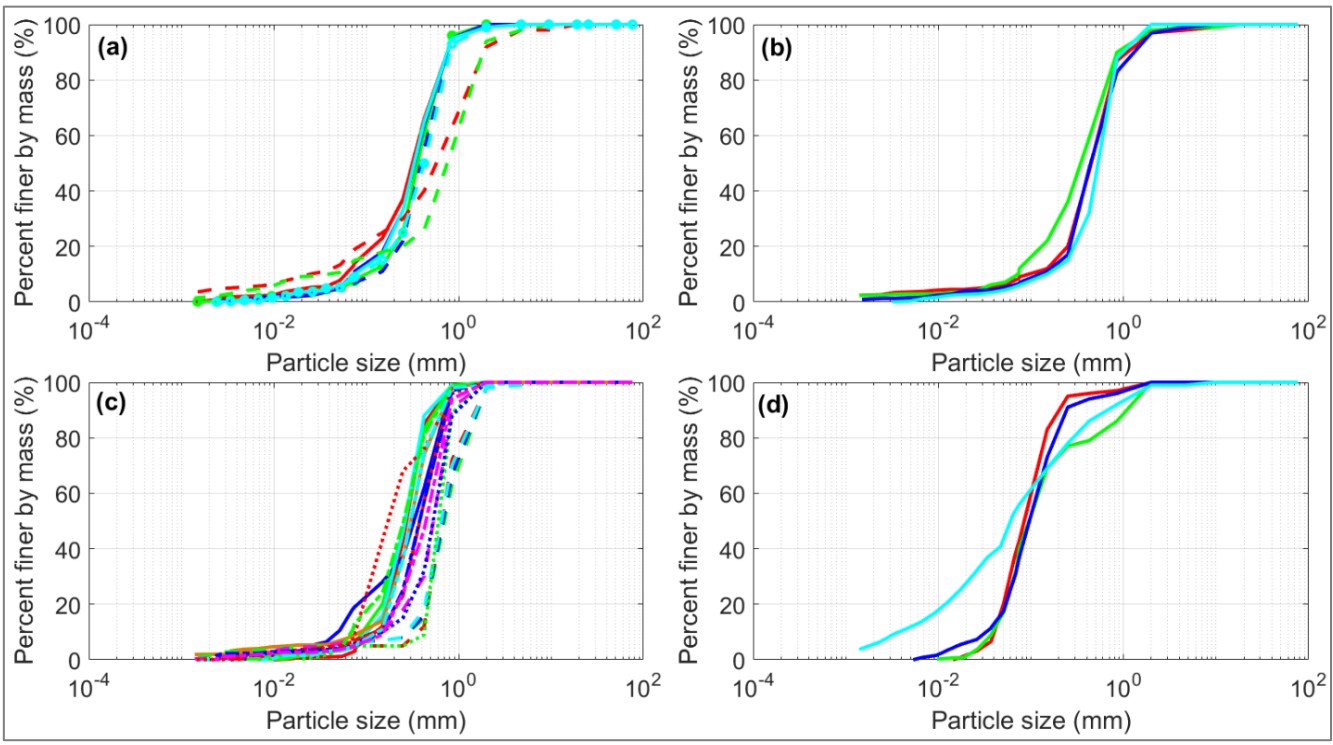

Figure 2: Particle size distribution (PSD) for (a) LFH, (b) peat, (c) subsoil, and (d) LOS materials for the treatment covers (OKC, 2009). The lines in the subplots show PSDs for different samples collected from the LFH, peat, subsoil, and LOS layers, respectively.

## 2.2 Field monitoring data

A climate monitoring station (Aurora Met) was established in 2012 to measure precipitation, air temperature, wind speed, net radiation, and relative humidity at the study site. The precipitation (rainfall and snow depth), air temperature, relative humidity, wind speed, and net radiation were measured daily and/or hourly using automated methods of measurement. The measurement instruments included a Texas Electronics TE525 tipping bucket (rain) and SR50 sonic ranging sensor (snow depth) for precipitation, a CS HMP45C sensor for air temperature and relative humidity, a RM Young 05103AP anemometer for wind speed, and a KIPP & Zonen NRLife net radiometer (OKC, 2017). Each treatment cell also had a soil monitoring location where volumetric water content, temperature, and suction were measured at multiple depths (typically every 10 cm) within the treatment covers and the underlying LOS. The volumetric water content was measured using Campbell Scientific CS616 Time Domain Reflectrometers (TDR), and the soil temperature and suction were measured using CS229 suction sensors (OKC, 2017). The monitoring data utilized in this study were collected from 2013 to 2016.

## 2.3 Parameter estimation using inverse modelling

The meteorological and soil monitoring data were used to calibrate a physically based SVAT model for each treatment cell based on IM. This provided a set of optimized soil hydraulic parameter values for the coversoil (LFH or peat), the subsoil, and the LOS. These parameters were interpreted to define spatial (cell to cell variation) and temporal (year to year variation) variability in the saturated hydraulic conductivity (Ks) and water retention curves (WRC). The Mualem tortuosity parameter was set to 0.5 and was not optimized as the goal was to only optimize a limited set of key parameters. This is denoted by l in Hydrus-1D and defined as the pore-connectivity parameter in the hydraulic conductivity function as estimated by Mualem (1976) to be approximately 0.5 as an average for many soils.

IM is a mathematical approach that estimates unknown causes (e.g., model parameters) using observed variables (e.g., water content and/or pressure heads) during a historical period by iteratively solving the governing equation (Hopmans et al., 2002). The governing equation (i.e., Richard's equation) for water flow in unsaturated soil was solved using HYDRUS-1D (Simunek et al., 2013). In HYDRUS-1D, potential evapotranspiration (PET) is calculated from climatic conditions using the Penman-Monteith equation (Brutsaert, 1982). It is then apportioned into potential evaporation (PE) and potential transpiration (PT) based on a prescribed leaf area index (LAI) value. The actual evaporation (AE) from the ground surface is calculated from the pressure head gradient between the top two nodes and hydraulic conductivity with two limiting conditions: (1) AE must be less than PE and (2) the calculated water pressure at the top node must be in the range from 0 kPa to a maximum suction equivalent to the atmospheric water vapor pressure. Actual transpiration (AT) is calculated by distributing PT over a prescribed rooting zone where root water uptake is limited by water stress, as calculated by a root water uptake model (Feddes et al., 1974). The root water uptake parameters were obtained from previous studies on the oil sands mine reclamation covers by Huang et al. (2011a, 2015, 2017). The Feddes model parameters were set as P0 = 0 kPa; P2H = -5000 kPa; P2L = -8000 kPa; P3 = -19000 kPa; r2H = 0.5 cm/day; and r2L =0.1 cm/day for all models as obtained from the preliminary study on the same sites by Huang et al. (2017).

HYDRUS-1D embeds an IM method into the numerical solution of the Richard's equation. The IM method uses the Marquardt-Levenberg gradient-based approach (Simunek et al., 2013) in which values of the five individual model parameters (i.e., $\theta r$, $\theta s$, $\alpha$, n, Ks) are varied for each material until a combination of the parameters is found that provides an optimal fit to the observed variation in a specific observation (i.e., volumetric water content) (Hopmans et al., 2002). The first four parameters ($\theta r$, $\theta s$, $\alpha$, n) are known as van Genuchten (VG) parameters (van Genuchten, 1980) and are used to describe the volumetric water content function (i.e., water content vs. suction). Ks is the saturated hydraulic conductivity of the soil. A closed form solution then estimates the hydraulic conductivity function (i.e., K vs. suction) from the VG parameters and Ks. How well these individual parameters are estimated determines the overall accuracy of parameter estimation. Details of IM used in HYDRUS-1D can be found in Simunek et al. (2013).

In this study, the embedded IM method in HYDRUS-1D was used to simulate volumetric water content by optimizing the soil hydraulic parameters until the simulated water content matched the measured values at various depths and times. To optimize

the parameters, an initial value as well as a search range defined by an upper and lower limit of each parameter were specified. The initial parameter values with their lower and upper limits for TRT 10 (Cell #23 in year 2013) are shown in Table 1 for the peat and subsoil reclamation materials as well as for the LOS substrate. To conduct IM in this study, the ranges of initial parameter values were estimated from the measured particle size distributions (PSDs) and bulk density using Arya-Paris model

(Arya et al. 1999). The water retention curves (WRC) for each PSDs from peat/LFH, subsoil, and LOS were estimated using the equations presented in the Arya-Paris model and the least-square optimization program RETC (van Genuchten et al. 1991) was used to fit the VG-Mualem equation to the estimated WRC from Arya-Paris model to estimate the VG parameters ($\theta_r$, $\theta_s$, $\alpha$, n). The Kozeny-Carman equation (Kozeny 1927; Carman 1938, 1956) was used to estimate Ks values from the PSDs as it is one of the most widely used and accepted methods (Huang et al. 2011a; Mathan et al. 1995). The estimation of parameters

using these methods helps to constrain the initial parameter ranges in the inverse modelling. In addition to the Ayra-Paris model, the initial range of $\theta_s$ can also be approximated from the measured water content data for the covers, where the maximum water content values are observed at the depths of 5-10 cm. After setting up the initial range of parameter values based on the above methods, the inverse modelling is repeated with different initial values. Once there is no significant change in $\theta_r$ and $\theta_s$ parameters and objective function (i.e. sum of least squares), these parameters are assumed optimized and kept

fixed in the subsequent IM for the remaining parameters. Step-by-step the least sensitive parameters are kept fixed and thereby reducing the number of parameters to be optimized by IM. Reducing the number of parameters, constraining the range of initial parameter values, and repeating the IM with initial parameter values were done as recommended by Hopmans et al. (2002). However, details of all these steps are not included in this manuscript, only referenced to Hopmans et al. (2002), for the brevity of the manuscript. It is important to note that the purpose of this manuscript was not to focus on IM techniques but rather to

highlight how reasonably optimized parameter sets can resemble the distribution of the measured key parameter (i.e. Ks) and represent the parameter variability. This comparison between the optimized and measured key parameter values was assumed an indirect validation of the inverse modelling approach used in this study, which can be used for further sampling based on PLHS with certain level of confidence.

Table 1: Initial value and lower and upper limits of the five soil hydraulic parameters for TRT 10 (Cell #23 in 2013) used in the inverse modelling to optimize parameters for peat, subsoil, and LOS

| | Parameter value | | | | |
|---|---|---|---|---|---|
| | *$\theta_r$ ($m^3/m^3$) | $\theta_s$ ($m^3/m^3$) | $\alpha$ ($m^{-1}$) | *n [-] | log(Ks) (m/s) |
| *PEAT* | | | | | |
| Initial value | 0.0160 | 0.610 | 4.50 | 1.50 | -4.53 |
| Lower limit | 0 | 0.461 | 2.50 | 1.20 | 0 |
| Upper limit | 0.0900 | 0.700 | 6.80 | 2.30 | -3.64 |

| | | | | | |
|---|---|---|---|---|---|
| *SUBSOIL* | | | | | |
| Initial value | -2.22 | 0.361 | 2.50 | 0.310 | -4.62 |
| Lower limit | 0 | 0.261 | 2.50 | 0.310 | 0.0 |
| Upper limit | -1.05 | 0.500 | 4.80 | 0.360 | -3.64 |
| *LOS* | | | | | |
| Initial value | 0.0900 | 0.368 | 4.50 | 1.74 | -6.14 |
| Lower limit | 0 | 0.328 | 3.00 | 1.55 | 0 |
| Upper limit | 0.0900 | 0.450 | 6.50 | 1.83 | -4.94 |

*The logarithmic (log10) values are shown for θr and n parameters of the subsoil layer

## 2.4 Discretization of model domain

The simulated model domain used in HYDRUS-1D had a maximum height of 2.50 m with a minimum of 1.00 m of LOS
overlain by the various soil profiles (Fig. 1b). The various soil cover designs (TRT) are summarized in Fig. 1b. Note the
following cover construction: Treatments 2 and 7-9 used LFH as the coversoil layer; Treatments 4, 8, and 10-11 were
constructed using blended B/C horizons as the subsoil; and Treatments 6, 9, and 12a were constructed using a Bm as the
subsoil layer. Figure 1b also demonstrates: choice of two depths (0.10 and 0.30 m) for the peat, two depths (0.10 and 0.20 m)
for the LFH, and various depths for the subsoil reclamation materials. The spatial discretization used for all of the model
domains was 1 cm and the time step was 86.4 s.

## 2.5 Initial and boundary conditions

Only the days in which the treatment covers were unfrozen were simulated in the IM. Snowmelt infiltration and drainage
following ground thaw were assumed to be complete prior to the start of the simulation. As a consequence, any snowmelt-
induced change in the soil water storage was already incorporated in the water content profiles from the first unfrozen day
(i.e., soil temperature > 0 °C). The measured volumetric water content profile of the first unfrozen day was set as the initial
condition, while a unit gradient (i.e. gravity gradient) was set as the lower boundary condition of the model domain. The SVAT
parameters (e.g., climate and vegetation characteristics) were used as the upper boundary condition.

## 2.6 Vegetation and root distribution

Maximum LAI values for each treatment cover were estimated from measurements by (Bockstette, 2017) and photographs
taken on site by OKC (2017). The estimated LAI values varied from 0.2 at TRT 5 to 1.5 at TRT 2, TRT 7, and TRT 8. Huang
et al. (2017) found that the temporal variation obtained with IM for similar sites was relatively minor compared to the spatial

variability in the cover properties. Examining the photographs revealed that the sites were initially bare and developed a vegetative cover over the first few years. Although the covers were planted with one of three tree species (i.e., trembling aspen, jack pine, white spruce) or a mix thereof, the dominant early establishment vegetation during the study period (2013-2016) was understory vegetation species (not trees). The understory development (i.e., density and species) was variable, depending on the treatment coversoil materials (i.e., peat or LFH; Jones, 2016). Due to the early dominance of understory species, the LAI was assumed to be relatively constant over the study period (i.e., 4 years). The seasonal distribution of LAI adopted for the simulations was the same as that used by Huang et al. (2015): (a) a linear rise in the spring from zero to a maximum value, (b) maximum in the summer, and (c) a linear decrease from the maximum value to zero in the fall.

In 2014, the maximum root depths used in the IM were 0.3 m at TRT 5; 0.5 m at TRT 1-4, TRT 6-8, TRT 10-11, TRT 12a, and TRT 12b; and 1.0 m at TRT 9 based on the measurements by Bockstette (2017). The roots were assumed to be distributed within the cover soils using an exponential function of root mass with depth, with the maximum root mass at the surface decreasing to zero at the maximum root depth. In the long-term simulations (discussed below), the root depths were assumed to have extended to the full depth of the covers.

### 2.7 Probability distributions of the optimized parameters

IM modelling was undertaken using the monitored water content profiles at each of the treatment cells along with the site-specific meteorological data for each individual monitoring year. Because one cell of TRT 5 was missing data in 2013, a total of 155 HYDRUS-1D models (13 treatments, 3 replicated cells, and 4 years of data) were calibrated by optimizing five soil hydraulic parameters for each soil type. The 155 sets of optimized parameters (both VG parameters and saturated hydraulic conductivity) were then used to populate a continuous probability distribution that represents the variability in each individual parameter. A cumulative density function (CDF) for each of the optimized parameters was plotted for all soil types to investigate if peat and LFH coversoil and all subsoil variations could be grouped together for the simplification of long-term water balance simulations.

The Kolmogorov-Smirnov (K-S) test was used to verify the distribution type of the five model parameters as obtained from the IM for all cells and all years. The K-S test checks the null hypothesis that a distribution belongs to a standard normal distribution (mean=0, standard deviation=1) if the resulting p-value is greater than the level of significance (e.g., 1%). The parameter values were centered and scaled using the corresponding mean and SD values prior to application of the K-S test. The distributions of the parameters that fail the normality check as stated above were log transformed, centered, and scaled before the K-S test. In addition, probability density functions were plotted for the parameters of each soil type to visually inspect the type of distributions.

## 2.8 Simulation of long-term water balance with parameter variability

### 2.8.1 Sampling of parameters

Alam et al. (2017) and Huang et al. (2017) used a limited number of alternate parameter sets to define variability to limit simulation times. The optimized soil properties (WRC and Ks) were grouped into discrete intervals representing the 10th, 25th, 50th, 75th, and 90th percentiles of the parameter distributions obtained from optimized parameter sets. The range of possible water balance outcomes (e.g., AT and NP) over a 60-year climate cycle was then simulated using the discrete percentiles parameter sets. However, these discrete (not randomly selected rather fixed) percentiles (i.e. 10th, 25th, 50th, 75th, and 90th percentiles) of parameter distributions are not representative of the whole range of parameter distributions.

A more efficient and sequential LHS-based sampling process—PLHS, as described above—was adopted in this study. According to Sheikholeslami and Razavi (2017), PLHS is an extension of conventional LHS, where PLHS consists of several sub-samples (called slices) in such a way that the union of these slices also retains the properties of the LHS. The PLHS sampling technique was implemented in this study using the MATLAB-based PLHS Toolbox developed by Sheikholeslami and Razavi (2017) to generate an appropriate sample size of n data points in a d-dimensional hypercube [0, 1] formed by the union of t small Latin hypercubes with m = n/t sample points. For example, an appropriate sample size is determined by generating a sample size of n parameter sets, where the maximum value of n was 2000 in a 5-dimensional (where 5 refers to the total number of parameters) hypercube formed by the union of 100 small Latin hypercubes. So, 20 sample sizes (equally sized slices) were obtained (i.e., m = 2000/20) to determine an appropriate sample size. Each of the 100 parameters sets was sequentially added to the next 100 parameter sets to generate PLHS-based parameter sets starting from 100 to 2000. While Hydrus-1D can be used to optimize parameters with reasonable computational cost, our goal was simply to use Hydrus-1D to optimize a set of parameters for each cover with each year's monitoring data. Thus, we obtained 155 sets of parameters which include 13 treatment covers, replicated in triplicate and monitored in four consecutive years. Since these parameters form a distributions of parameters representative of the measured parameter distributions (at least for Ks), we decided to use a standard sampling technique (e.g. PLHS) to do the rest with regards to generating multiple sets of parameters. Comparison between the multiple sampling from Hydrus-1D and from PLHS could be an extended study of the current in terms of both performance and computational cost.

Once the distributions of both the optimized and PLHS-based sampled parameters were verified to be similar, the appropriate number of randomly sampled parameter sets was used to simulate realizations of AET and NP over 60 years of climate variability using HYDRUS-1D. The realizations of AET and NP were expected to encompass a wider range of variability in the water balance of the reclamation covers due to parameter variability than using discrete percentiles of the optimized parameters. The classical MC sampling method was also used to verify its limitations relative to the PLHS and discrete sampling approaches.

### 2.8.2 Illustrative covers

The long-term cover performance was evaluated by simulating long-term climate records represented by 62 years (1952-2013) of climate data from Fort McMurray Airport Weather Station. The first two years (1952-1953) were used to spin up the model and establish the initial conditions. Variability in the long-term cover performance was incorporated by simulating five illustrative covers of 0.20-m peat and 1-m LOS overburden with five different depths of subsoil [A50 (0.50 m), A75 (0.75 m), A100 (1.00 m), A125 (1.25 m), and A150 (1.50 m)] with the PLHS-based sampled soil properties. Similar illustrative cover designs were used in Alam et al. (2017) and Huang et al. (2017) with minor modifications in the model domain, where the order of the soil profile was peat/subsoil/LOS.

The modelling approach (model domain, spatial/temporal discretization, etc.) was the same as for the IM modelling but with several key differences. First, the accumulated snowpack from winter precipitation was added to the cover in the early spring of each year. While runoff from the watershed would largely depend on the slope of the watershed, the amount of runoff would vary between the reclamation cover systems. Huang et al. (2015) showed an average runoff of 34 mm each year from a sloping cover (~5H:1V), while other reclamation covers were flat-lying and assumed to have negligible runoff in previous studies (Alam et al. 2018; Huang et al. 2015, 2017). So, the runoff from the flat-lying reclamation cover was not simulated in this study rather incorporated in the NP rates. Therefore, the simulated NP rates represent the total water yield from the covers that may eventually reach the downgradient surface water bodies. Besides, there was no measurement to confirm which one between runoff and infiltration dominates in the reclamation cover sites. The melt volume was calculated using the degree-day method (Carrera-Hernández et al., 2011) when the mean daily temperature was greater than 0 °C, and was then added to any precipitation occurring during the winter period and to any stored water in the soil profile in the early spring of each year. This method of calculating melt volume uses a constant that accounts for all the factors affecting the snow melt amount and varies with time. The method did not consider sublimation as intercepted snow results in the highest rates of sublimation; however, interception of snow is quite low in case of a deciduous tree (e.g. aspen). Second, the roots were assumed to have an exponential root distribution that fully penetrates the covers without penetrating into the LOS layer. It is possible that the roots would eventually penetrate into the LOS substrate over the long-term period; however, this more conservative assumption restricts root water uptake to the reclamation materials. The maximum root depth assumed in this study seems reasonable compared to the root depths of tree species, between 3 and 57 years of age, in boreal forests (range 0.3 to 2 m; Strong and La Roi, 1983). Third, the method proposed by Huang et al. (2011b, 2017) was used to constrain the LAI values used in the simulation based on the predicted range of AET values. The maximum sustainable LAI (LAI_max) values were evaluated to ensure the predicted values of AET were sufficient to support the prescribed LAI used in the simulations. In the IM, the measured LAI values were used to obtain the optimized model parameters, and no significant evolution in the LAI values was observed or simulated. However, the long-term simulation of water balance requires a specified pattern of seasonal variations in LAI to determine the LAI_max for each illustrative cover. The seasonal variations in LAI were represented in a similar way to Huang et al. (2015) using six seasonal patterns of LAI (i.e., LAI of 1, 2, 3, 4, 5, and 6) for each illustrative cover. Huang et al. (2011b, 2015) and

Alam et al. (2018a) used literature-based relationships between above-ground net primary production (ANPP), LAI, and actual evapotranspiration (AET) to constrain LAI_max values in the long-term simulations. Because parameter variability is expected to influence the long-term water balance (AET and NP) of the treatment covers, the ANPP-LAI-AET relationships are also expected to be influenced by the parameter variability. Consequently, the variability in LAI_max has an influence on the long-term cover performance in combination with the parameter variability. For details of this approach, interested readers are referred to Alam et al. (2018b).

## 2.9 Statistical methods

The K-S test was used to verify the distribution of the optimized parameter values. The mean and SD were used as the convergence criteria while selecting an appropriate sample size. The PLHS method was used to sample from the distributions of the VG parameters and Ks using various sample sizes between 15 and 2000. When the mean and SD of the sampled parameters converge to the mean and SD of the optimized parameters and remain unchanged, the sample size was considered "appropriate". The uniformly distributed sample points in the PLHS approach were transformed to a normal distribution using the inverse cumulative distribution function (i.e., ICDF as a transfer function). The parameters showing log-normal distribution were transferred to a normal distribution using log transformation prior to using the ICDF.

## 3 Results and discussion

### 3.1 Performance of inverse modelling for the treatment covers

The performance of the inverse modelling technique of Hydrus-1D model was first evaluated by comparing the measured and simulated water contents at various depths within each of 13 treatment covers. The coefficient of determination ($R^2$) and root-mean-square errors (RMSE) between the measured and simulated water contents are shown in Table 2, while the comparison between the measured and simulated water contents at various depths within each of 13 treatment covers in a typical year during 2013-2016 is shown in Fig. 3. For the treatment covers, the $R^2$ values are mostly above 0.8, and RMSE values are mostly less than 1 mm/day, except for a few treatment covers. The performance criteria as well the graphical comparison between the measured and simulated water contents at various depths within the treatment covers show that the models perform reasonably well given diverse soil conditions, number of treatment covers, and number of parameters to be optimized.

Table 2: Performance statistics ($R^2$ and RMSE) of inverse modelling for each of 13 treatments covers at the Aurora North Mine site

| Treatment cover # | $R^2$ | RMSE (mm/day) |
|---|---|---|
| 1 | 0.89 | 0.66 |
| 2 | 0.82 | 0.57 |
| 3 | 0.73 | 0.40 |
| 4 | 0.81 | 0.79 |
| 5 | 0.62 | 1.00 |
| 6 | 0.86 | 1.07 |
| 7 | 0.79 | 0.34 |
| 8 | 0.82 | 0.39 |
| 9 | 0.51 | 1.06 |
| 10 | 0.84 | 0.72 |
| 11 | 0.84 | 0.71 |
| 12a | 0.81 | 0.28 |

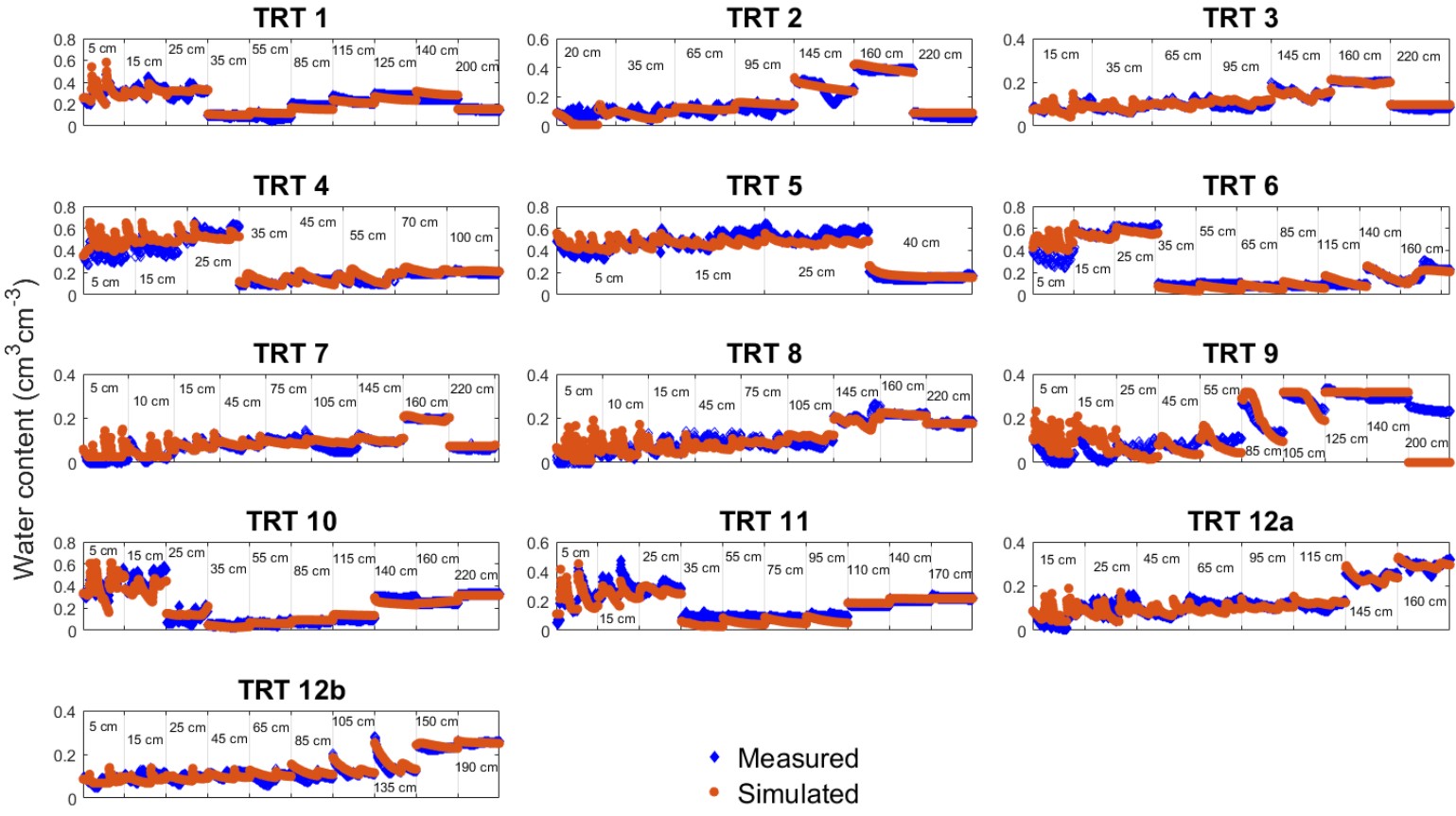

Figure 3: Comparison between the measured and simulated water contents at different depths within each of 13 treatment covers for the days when temperature is greater than 0 oC. Typical depths at which the water content measurements are recorded vary from 5 to 200 cm within the treatment covers.

## 3.2 Probability distributions of the optimized and sampled parameters

The K-S test was used to verify the distributions for each of the five parameters from IM of all cells and all years. K-S test results indicate that the VG parameters of soil types (except θr and n of subsoil) were normally distributed at the 1% significance level. The θr and n parameters of subsoil were log-normally distributed at the 0.001 and 0.1% significance levels, respectively. Ks was log-normally distributed (Fig. 4) at the 1% significance level. Ks values are commonly found to be log-normally distributed (Huang et al., 2017; Kosugi, 1996).

Despite differences in the CDF (see Fig. A1) of the optimized parameters for the peat or LFH coversoil as well as differences between various salvaged subsoils and different LOS overburden materials the treatment cover materials were grouped as peat, subsoil, and LOS. This grouping was adopted for the purposes of this study because it maximizes the number of IM parameter sets and helps illustrate the impacts of parameter uncertainty on expected performance. According to Syncrude Canada Ltd., in the final cover design the top layer might be either peat/LFH or combination of the two. The distributions of parameters for these two materials together seem reasonable to be used in the illustrative covers for long-term simulation of water balance components. Moreover, the primary purpose of this study was not to differentiate the performances of two alternate coversoils built on the two organic-rich materials. Therefore, the PLHS method was used to randomly sample from the distributions of the two materials grouped together and the distributions of parameters for these two materials together are used in the illustrative covers for long-term simulation of water balance components.

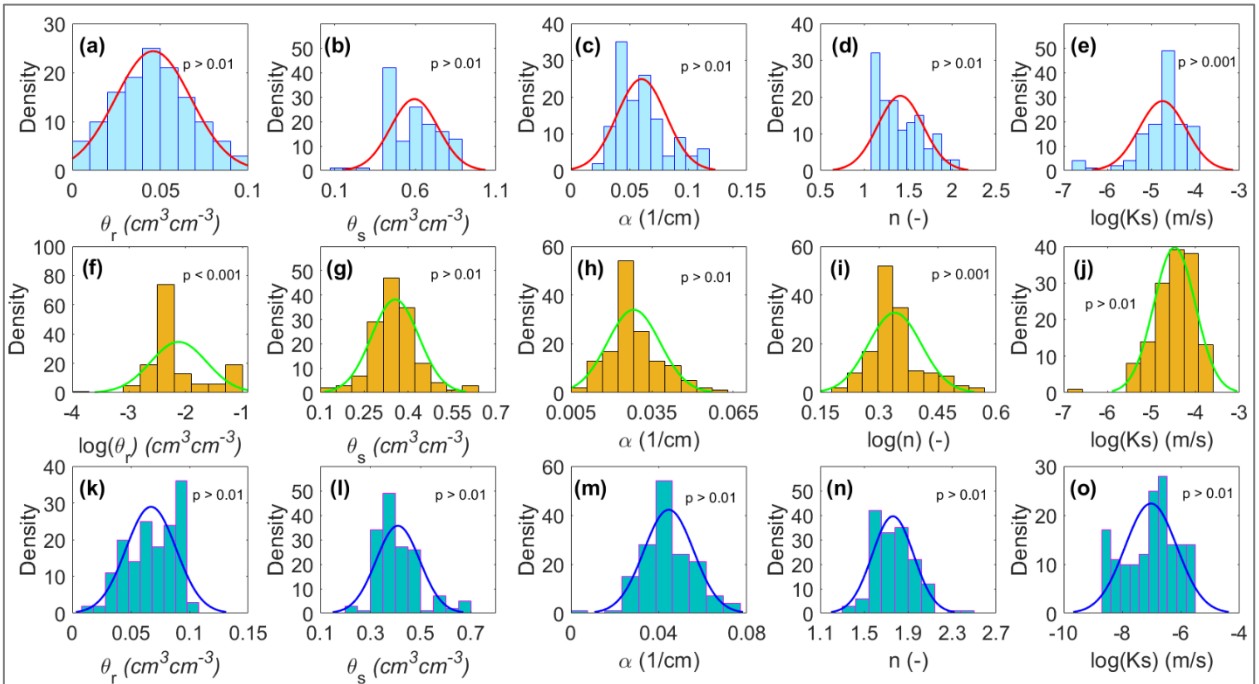

Figure 4: Probability density functions (PDFs) fitted to the four VG parameters and Ks obtained from the IM for the three soil types: peat (a-e), subsoil (f-j), and LOS (k-o). The mean and SD of the fitted distributions are shown in Table 2.

5    The variability in the optimized parameter values includes both spatial and temporal variability. The material properties of the treatment covers evolve with time as they vary in space. It seems important to see how these material properties would vary in time, if any, in addition to the spatial variability. The probability density functions (PDFs) of Ks obtained for all of the cells and all years represent the total variability (spatial plus temporal) in the Ks parameter while the PDFs for each treatment cell averaged over the four monitoring years represent spatial variability alone. Comparison of these PDFs (Fig. 5) for Ks (only

10   for Ks as it was the most influential parameter for the treatment covers) shows that the spatial variability contributed more than 90% (as 90% of the PDF corresponding to spatial variability falls within the PDF corresponding to total variability) of the total variability for the three materials. Because temporal variability was not significantly contributing to the total variability, the spatial and temporal variabilities were not separated from the total variability in this study.

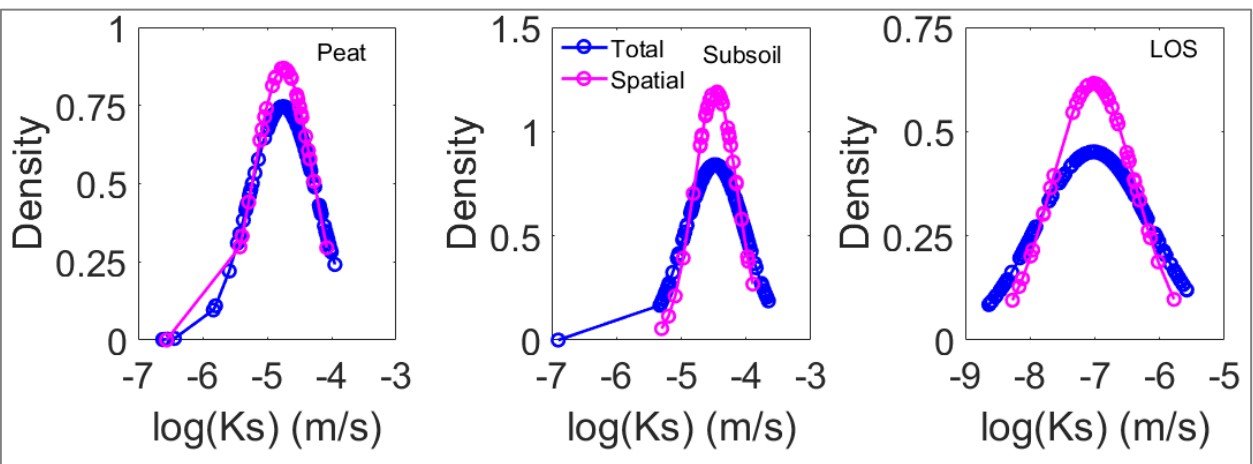

Figure 5: Probability density functions (PDFs) of Ks total variability and spatial variability for the three materials.

A total of 700 parameter sets were randomly sampled from the prescribed probability distributions using the PLHS method. These sampled distributions (Fig. 6) accurately captured the IM parameter distributions with the exception of the residual water content ($\theta_r$) for the subsoil layer. In this case, a large number of the IM values were close to zero and the randomly sampled $\theta_r$ values consequently underestimate the optimized $\theta_r$ values.

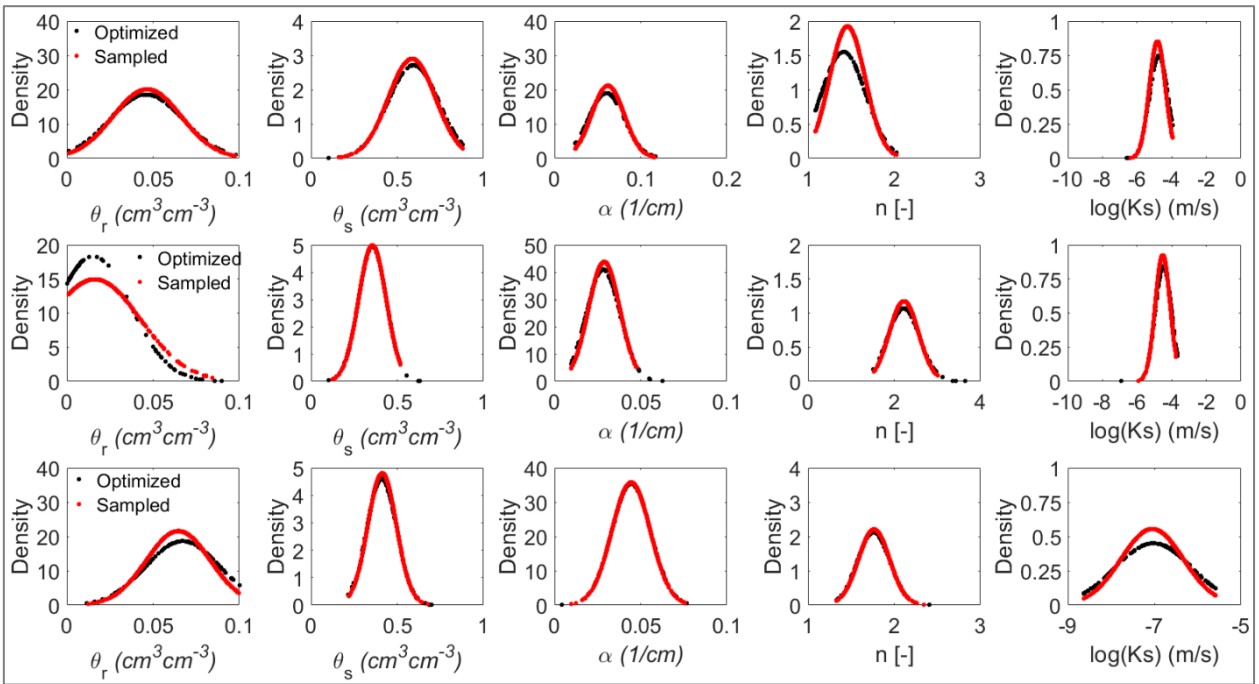

Figure 6: Probability density functions (PDFs) for the optimized and sampled 700 parameter sets for the peat coversoil (top row), subsoil (middle row), and LOS (bottom row).

A further comparison between the optimized and sampled parameter values in terms of their basic statistics (e.g., mean and SD) is shown in Table 3. The percentage difference between the mean of the sampled and optimized parameter values varies between 0.01 and 5.47%. The average error of 1.64% includes the larger errors associated with $\theta r$ for the subsoil. The percentage difference between the SD of the sampled and optimized parameter values varies between 0.21 and 24.72% with an average error of 8.29%, including the errors in approximating $\theta r$ for the subsoil and Ks of LOS. The larger error in the approximation of subsoil $\theta r$ and LOS Ks is related to overestimation of the optimized $\theta r$ values and underestimation of the optimized Ks values by PLHS sampling. Overall, the random sampling approach seems to provide a good approximation of soil hydraulic parameters with regards to their mean and SD values as well as the corresponding PDF patterns.

Table 3: Mean and SD values of the optimized and randomly sampled parameters for peat, subsoil, and LOS. The difference between the corresponding mean and SD of the sampled and optimized parameter values are shown as percentages.

| Parameter | Peat | | | Subsoil | | | LOS | | |
|---|---|---|---|---|---|---|---|---|---|
| | Optimized | Sampled | Error (%) | Optimized | Sampled | Error (%) | Optimized | Sampled | Error (%) |
| $\theta r$ ($m^3m^{-3}$): Mean | 0.0460 | 0.0470 | 1.11 | 0.0150 | 0.0160 | 5.47 | 0.0670 | 0.0650 | 3.69 |
| $\theta r$ ($m^3m^{-3}$): SD | 0.0210 | 0.0200 | 7.54 | 0.0217 | 0.0161 | 24.7 | 0.0210 | 0.0190 | 13.6 |
| $\theta s$ ($m^3m^{-3}$): Mean | 0.594 | 0.585 | 1.47 | 0.356 | 0.356 | 0.0100 | 0.410 | 0.412 | 0.670 |
| $\theta s$ ($m^3m^{-3}$): SD | 0.147 | 0.137 | 6.32 | 0.0810 | 0.0800 | 0.950 | 0.0870 | 0.0820 | 4.13 |
| $\alpha$ ($m^{-1}$): Mean | 0.0600 | 0.0620 | 2.98 | 0.0280 | 0.0290 | 2.28 | 0.0450 | 0.0440 | 0.160 |
| $\alpha$ ($m^{-1}$): SD | 0.0210 | 0.0190 | 10.2 | 0.0100 | 0.0100 | 1.35 | 0.0110 | 0.0110 | 0.790 |
| n [-]: Mean | 1.41 | 1.46 | 3.22 | 2.22 | 2.23 | 0.230 | 1.76 | 1.77 | 0.280 |
| n [-]: SD | 0.257 | 0.208 | 19.2 | 0.0373 | 0.348 | 0.820 | 0.187 | 0.184 | 3.47 |
| log(Ks) (m/s): Mean | -4.75 | -4.85 | 1.53 | -4.48 | -4.52 | 0.970 | -7.02 | -7.05 | 0.420 |
| log(Ks) (m/s): SD | 0.534 | 0.468 | 12.4 | 0.430 | 0.431 | 0.210 | 0.884 | 0.719 | 18.6 |

### 3.3 Distribution of WRC and Ks parameters

The WRCs for the three cover soils were defined by the four IM-generated VG parameters. These individual parameter distributions were randomly sampled 700 times to generate 700 WRCs. The various VG parameters were considered as independent parameters with no correlation between them. The choice of 700 samples was selected from sampling tests described below. The 10th percentile, mean, and 90th percentile of the 700 calculated WRCs based on the 700 randomly sampled sets of VG parameters were compared to the corresponding WRCs obtained from the 155 IM-based parameter sets (Fig. 7). This comparison is not intended to be a validation of the sampling approach but more of a visual comparison of the 155 WRCs generated from IM with 'virtual' WRCs generated from random sampling of the individual WRC parameters. Despite the correlation between these parameters in the form of a water retention curve (WRC), the PLHS method randomly selected these parameters without considering the correlation between them. However, the PLHS method was able to maintain those correlations when plotted as WRCs as shown in Fig. 7 and turns out to be a reliable method that captures the physical relationship between the VG parameters.

The distribution of WRCs is represented by the mean and 90% confidence intervals (CIs) of WRCs based on the 155 optimized VG parameters (i.e., $\theta_r$, $\theta_s$, $\alpha$, n) compared to the WRCs generated based on the 700 sampled VG parameters. The randomly sampled VG parameters provide a good representation of the optimized WRCs with a $R^2$ value of 0.99 for all soil types but with some visually apparent discrepancies in the tails. Generally, the extreme values belong to one of three distributions (Gumbel, Fréchet, or Weibull); however, the PLHS-based sampling was performed using normal distributions of the optimized VG parameters.

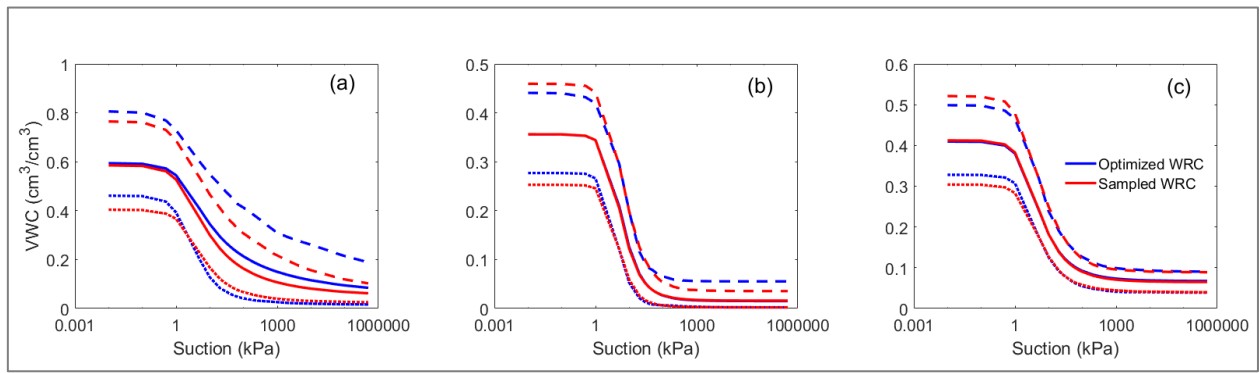

Figure 7: Estimated mean (solid lines) with 10th (dotted lines) and 90th (dashed lines) percentiles of the soil water retention curves (WRCs) for (a) peat, (b) subsoil, and (c) LOS obtained from the 155 optimized and 700 randomly sampled parameter values, where VWC denotes volumetric water content.

Huang et al. (2016, 2017) note that the cumulative frequency distributions (CFDs) for Ks values obtained from IM (i.e., optimized Ks values) were similar to those obtained from direct field testing. The field Ks values were measured using air permeameter (AP) and Guelph permeameter (GP) testing. Huang et al. (2016) show that the Ks values from AP and GP testing produced very similar descriptions of variability, although the mean Ks values were slightly offset as might be expected. The sampled Ks values are compared to the optimized Ks values and Ks values obtained from direct field measurements in Fig. 8. The CFD of the Ks obtained by random sampling produces a similar distribution to the IM distribution. Similar to the comparisons for the WRC, the random sampling exhibited more 'tailing' at the lower values of Ks for the peat and subsoil while creating a much smoother distribution than those obtained by the optimized Ks values. The discrepancy between the optimized and sampled Ks distributions derives primarily from sampling of the log normal distribution that was fit to the IM distribution. This ensures that the statistical characteristics of the distribution are retained and reflected in the sampled distribution but may result in specific deviations from the modelled (IM) distribution.

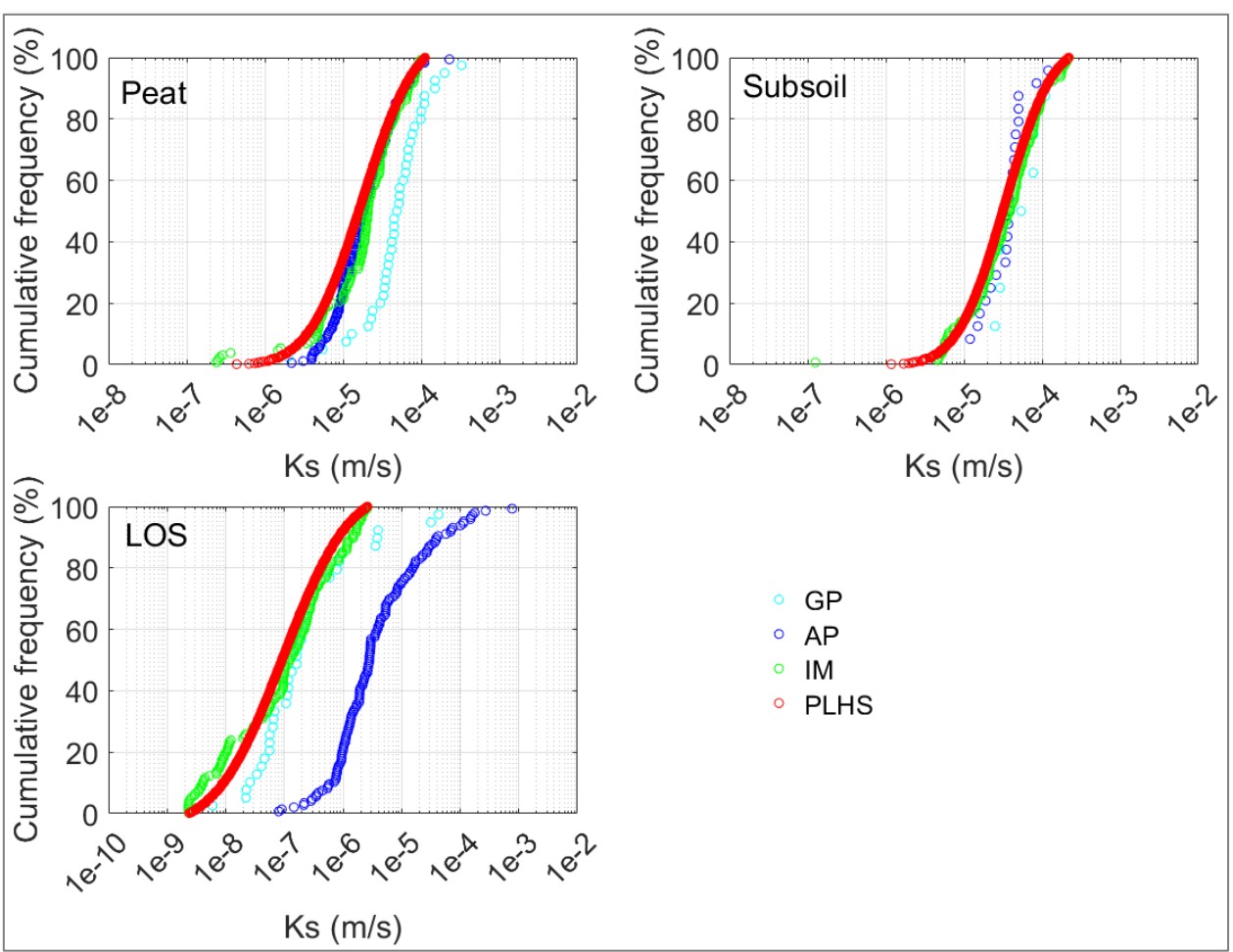

Figure 8: Comparison of the cumulative frequency distributions of the field-measured Ks using GP and AP methods, and optimized (IM) Ks values with the randomly sampled (PLHS) 700 Ks values. The results shown are for peat, subsoil, and LOS soil types.

A key parameter of Hydrsu-1D model for simulating water balance components at the reclaimed land has been the saturated hydraulic conductivity (Ks), which has been measured in the field using couple of methods. While Ks influences the net percolation rates, the root distribution influences the transpiration rates from the plants on the reclamation covers. The root water uptake model by Feddes et al. (1974) was used in this study, where the root distribution was approximated using exponential equations showing the relationship between relative root density and depth for the treatment covers since exponential root distribution was found to perform better in the near surface horizons (Li et al. 1998). However, the root distribution is affected by site conditions (Strong and La Roi 1983). Different root distributions, e.g. exponential, combination of uniform and exponential, and linear etc, were obtained from previous studies (Alam et al. 2018; Huang et al. 2011c, 2015) and evaluated in this study. Finally, the exponential root distributions seem to produce parameter sets where the distributions of Ks match reasonably well with the distributions of measured Ks values.

### 3.4 Selection of an appropriate sample size for PLHS

The probability distributions of the five optimized parameters were sampled using 26 different PLHS sample sizes (ranging from 15 to 2000) and the mean and SD for each sample set were calculated (Fig. 9). The mean and SD values clearly converge and remain relatively unchanged when the sample size exceeds 500 in most cases. Comparable sampling sets using the MC method would require more than 5000 samples to reach a similar level of convergence (Figs. A3 and A4 in Appendix A). To keep the simulation time reasonable, a sample size of 1000 was used to simulate water balance components for the illustrative covers.

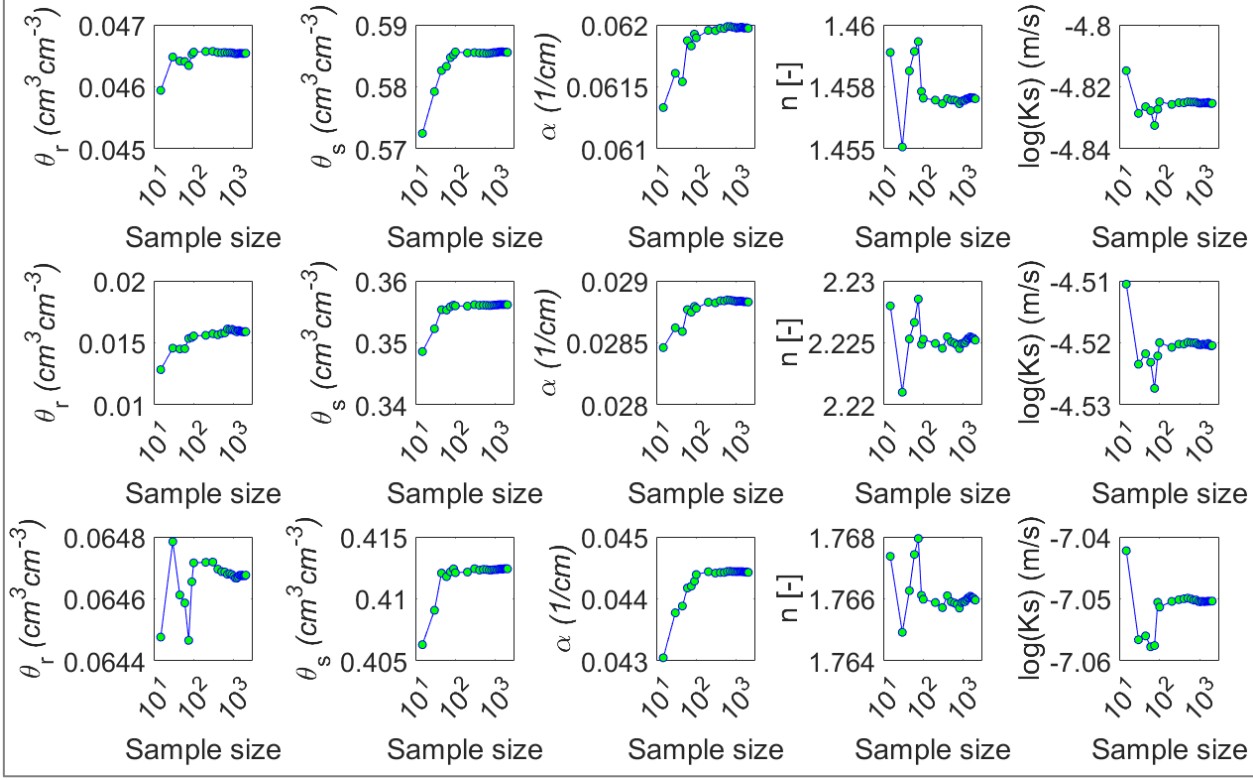

b.

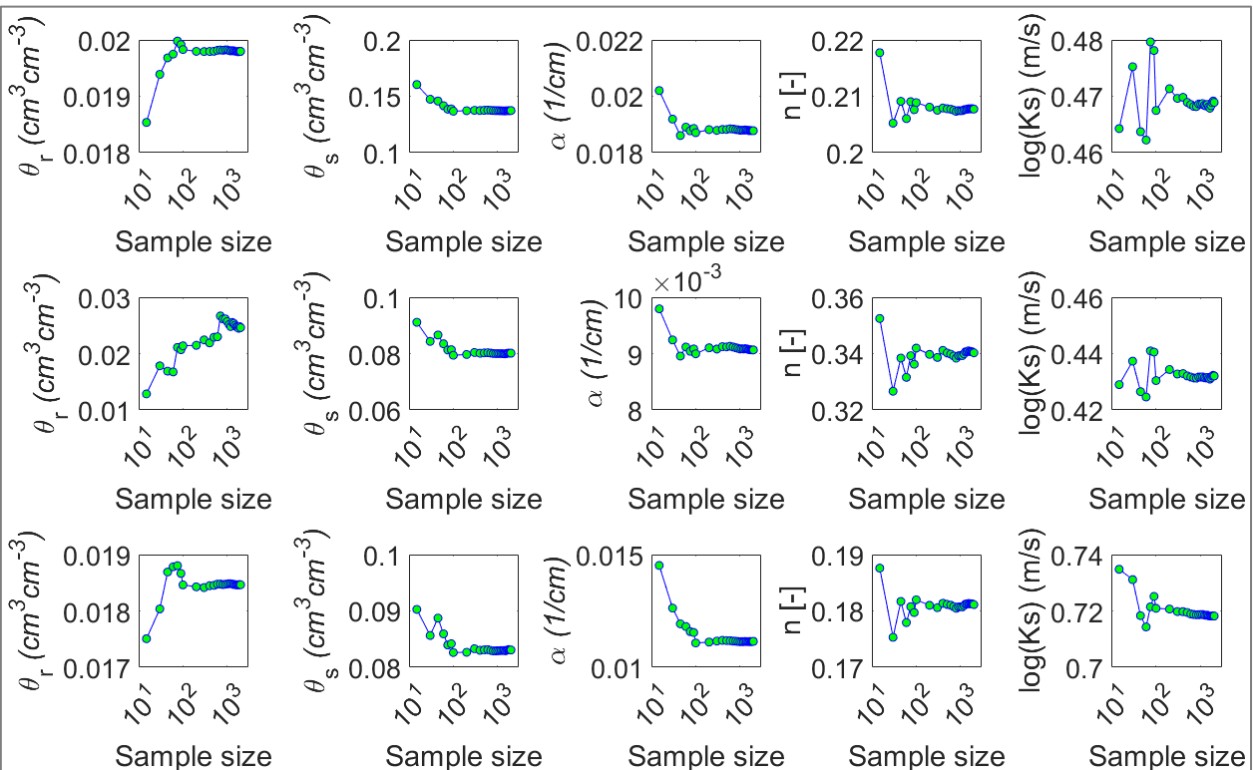

Figure 9: (a) Mean and (b) SD of the sampled parameter values corresponding to each sample size. Results are shown for peat (top row), subsoil (middle row), and LOS (bottom row) in both (a) and (b).

The impact of the varying PLHS sample sizes on water balance outcomes (i.e., AET and NP) was also evaluated. A set of 16 different sample sizes (i.e., 15 to 1000) was used to simulate AET and NP (i.e., using a total of 5815 simulations over a 60-year climate cycle) for the A100 illustrative cover (i.e., 0.20 m of peat and 1 m of subsoil placed over 1 m of LOS) with an LAI of 3.0. The results (Fig. A2 in Appendix A) show that the mean and SD of the AET and NP values also converge when the sample size is larger than 500. To be conservative, a PLHS sample size of 700 was used to define the hydraulic parameter distributions for the long-term simulation of the illustrative covers. However, a sample size of several hundred might also have been chosen.

### 3.5 Determination of maximum sustainable LAI using sampled parameter sets

The variability in LAI_max was evaluated using the lower bound (10%), mean, and upper bound (90%) of the simulated annual AET values (Fig. 10) for a series of simulations in which the LAI was set to one of six values (i.e., 1.0, 2.0, 3.0, 4.0, 5.0, and 6.0). A literature-based line representing the annual AET required to support a particular LAI value was also plotted on this

figure. The intersection points between the simulated and required AET lines were designated as the LAI_max values for each of the five covers. The mean LAI_max values range from 4.12 to 4.50 for the five illustrative covers as shown in Fig. 10. The respective lower, mean, and upper LAI_max values for each cover were as follows: A50 (2.73, 4.12, and 5.23); A75 (2.79, 4.25, and 5.36); A100 (2.86, 4.27, and 5.42); A125 (2.94, 4.37, and 5.53); and A150 (3.06, 4.50, and 5.68). These results indicate that all LAI values increase with increasing cover thickness but the difference between the lower and upper LAI_max values also increases with cover thickness.

Huang et al. (2015) showed that the increases in AET are not necessarily proportional to the incremental increases in cover thickness, rather little increment is noticed in the median AET over a climate cycle once a threshold cover thickness is passed. Therefore, it is not a surprise to observe the narrow range of LAI_max values as shown in Fig. 10. That said, there is support for decreased NP rates for thicker covers as greater volumes of water can be stored and ultimately released as AET.

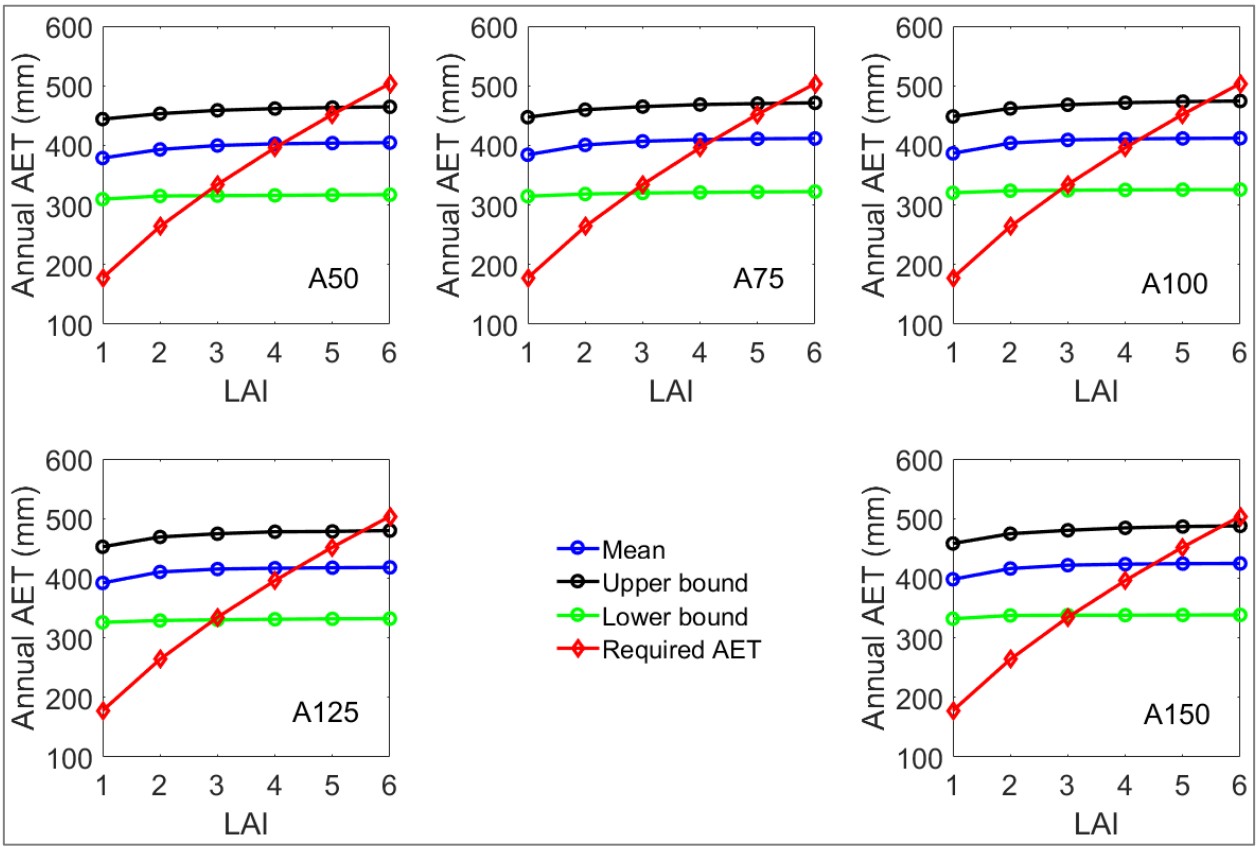

Figure 10: Lower (10%), mean, and upper (90%) limits of LAI_max values for the five illustrative covers showing variability in the LAI values based on the simulated annual AET with the 700 parameter sets over a 60-year climate cycle.

### 3.5.1 Uncertainty in determining the LAI_max values

The LAI_max values (i.e., the mean LAI_max values) from Fig. 10 for each illustrative cover were used to simulate the annual AET values for the 60 years of climate data. The results shown in Fig. 10 combine the impact of both climate variability and parameter uncertainty (700 parameter sets) on the relationship between LAI_max and the major water balance components of NP and AET. Table 4 (last column) shows the mean of LAI_max values as well as the corresponding standard deviations (SD) as calculated from the simulated AET values for the five illustrative covers. The mean and SD of the LAI_max values demonstrate that the parameter variability results in slightly higher LAI_max and slightly lower uncertainty as cover thickness increases. The mean LAI_max values were found to be around 4.12 to 4.50 considering all cases. Overall, the SD of LAI_max values ranges from 3.80 to 4.70 for all five illustrative covers depending on whether the climate year is drier or wetter. The range is within the measured LAI range for the Canadian boreal forest shown by Barr et al. (2012) to be between 2.0 and 5.20 based on old aspen, old black spruce, and old jack pine.

### 3.6 Uncertainty in estimating water balance components

### 3.6.1 Impact of parameter, climate, and LAI variability on the simulated AET and NP

Three primary sources of variability in the simulated AET are parameter uncertainty, climate variability, and LAI variability. Fig. 11 shows the distributions of AET resulting from these sources of variability, which were obtained from the simulated AET using 700 parameter sample sets, 60 years of climate data, and three LAI_max values (i.e., min, mean, max). The impact of the three sources of variability was separated as follows: (a) the simulated annual AET values corresponding to the mean LAI_max were averaged over the 60-year climate cycle to demonstrate the parameter uncertainty; (b) the simulated annual AET values corresponding to the mean LAI_max were averaged over the 700 parameter sets to demonstrate the impact of climate variability; and (c) the simulated annual AET values corresponding to the min, mean, and max LAI_max were averaged over the 60-year climate cycle and 700 parameter sets to demonstrate the LAI variability.

The AET distributions are shown as box plots for the five illustrative covers in Fig. 11. The results demonstrate that the variability in the simulated AET derived from parameter uncertainty decreases slightly with increasing cover thickness while remaining relatively constant with cover thickness in the case of climate variability. The median annual AET values resulting from the parameter variability and climate variability distributions are similar, particularly for the thinner illustrative covers. Overall, climate variability exerts more impact on the simulated annual AET followed in turn by parameter variability and then LAI values.

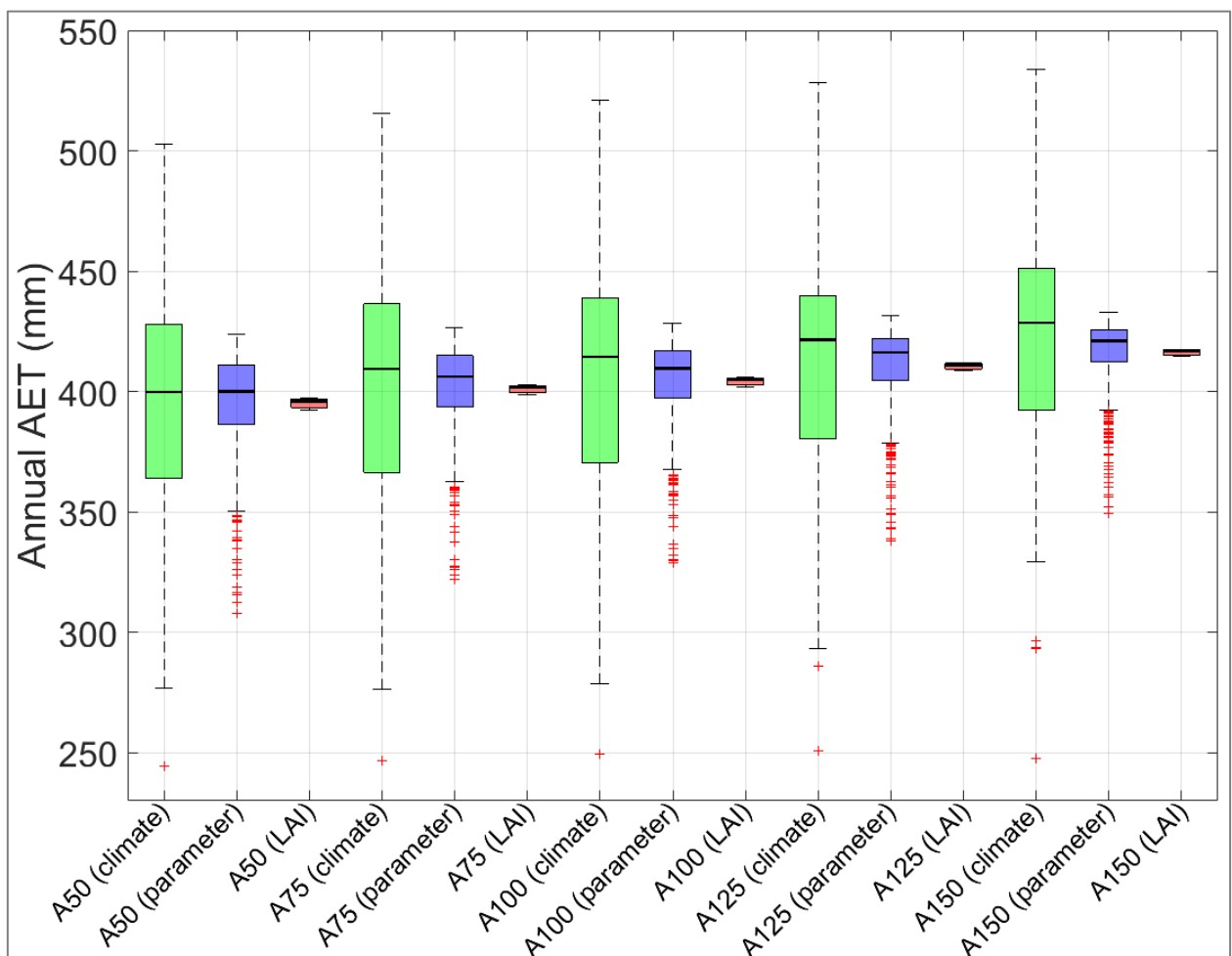

Figure 11: Box plots showing the distributions of annual AET obtained from the simulated water balances for the five illustrative covers with 700 parameter sets (blue boxes show parameter variability) over a 60-year climate cycle (green boxes show climate variability) with four LAI values (red boxes show LAI variability). The heavy dark line in each box plot shows the median, the boxes show the first and the third quartiles, and the whiskers extend to 1.5 times the inter-quartile range. Outliers are shown as red plus signs.

The impact of the three sources of variability on annual NP is presented in Fig. 12. The NP distributions for the five illustrative covers demonstrate that the variability in simulating NP decreases with increasing cover thickness for both the parameter and climate variability cases. In contrast to the AET results, the variability associated with climate variability is similar to that obtained by parameter variability. The distance between the median NP and the first quartile as well as the length of the whiskers seem to decrease with increasing cover thickness, which would mean less extreme annual NP for the thicker covers. The difference between the median annual NP values obtained from the parameter variability and climate variability appears

to decrease with increasing cover thickness. Overall, the parameter uncertainty and climate variability have similar levels of influence on the simulated annual NP, followed by the variability due to the calculated LAI values.

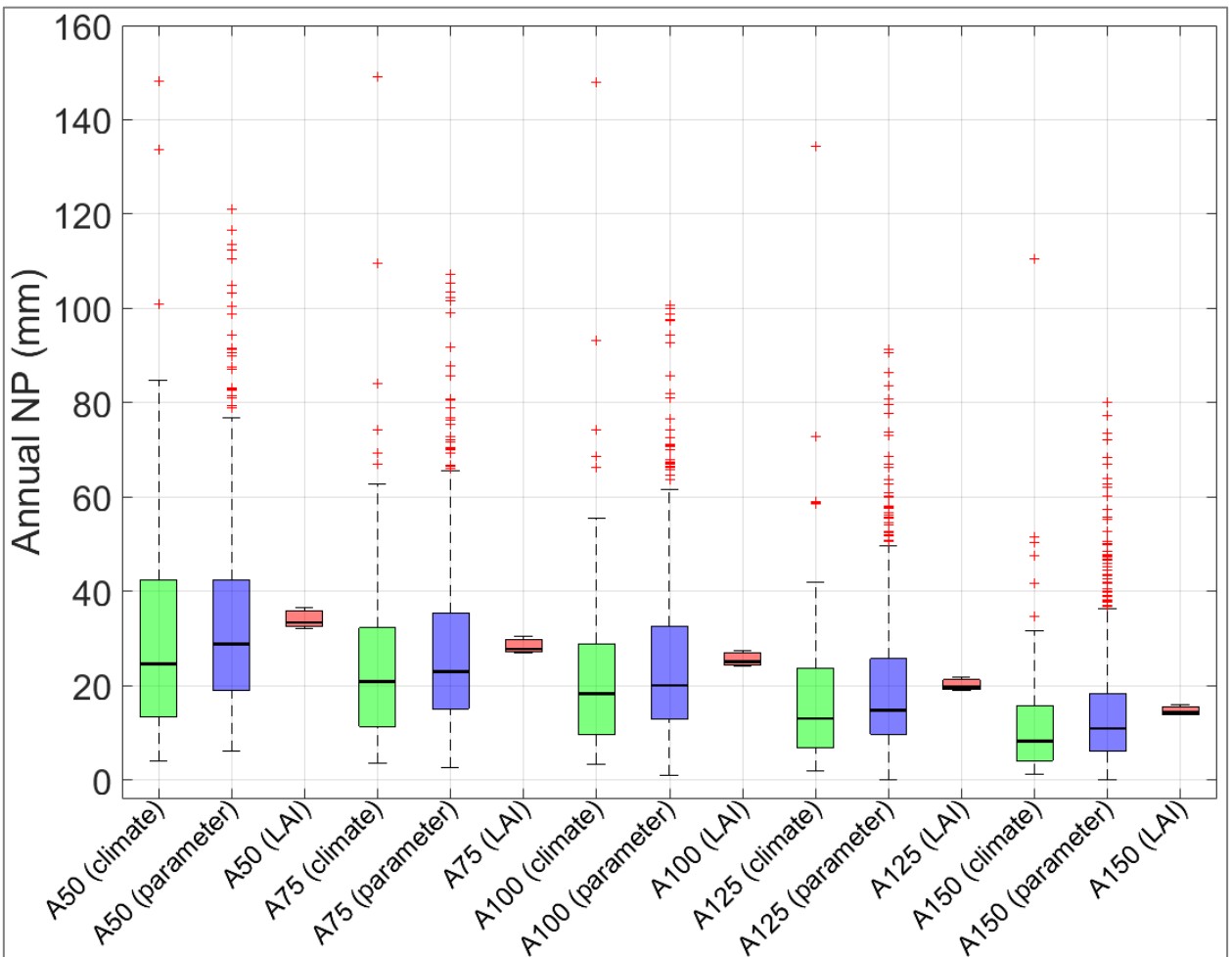

Figure 12: Box plots showing the distributions of annual NP obtained from the simulated water balances for the five illustrative covers with 700 parameter sets (blue boxes show parameter variability) over a 60-year climate cycle (green boxes show climate variability) with four LAI values (red boxes show LAI variability). Description of the box plots is the same as for Fig. 11.

The water balance components for the five illustrative covers are summarized in Table 4 based on the corresponding mean LAI_max values for each cover. Among the five illustrative covers, A50 had the lowest annual AET and the highest annual NP. The annual AET increases with increasing cover thickness, whereas the annual NP decreases with increasing cover depth, similar to previous studies (Huang et al., 2015; Alam et al., 2018a). These trends of annual AET and NP are consistent with the LAI_max values for each soil cover: higher annual AET for the covers with higher LAI_max values and higher annual NP for the covers with lower LAI_max values. The relative AET values reflect the relative productivity of the five illustrative

covers, while the relative NP values indicate the thicker covers are capable of storing more water than the thinner covers. Table 4 also shows the annual soil moisture deficit (DS) values for the five illustrative covers, where DS increases with increasing cover thickness.

The results showed that climate variability is a key source of uncertainty for the simulated AET during the historical 60-year period. That said, climate and parameter variabilities appear to cause similar levels of uncertainty in the simulated NP rates during the same historical period. Our previous studies (Alam et al. 2017b, 2018a) showed that the median AET and NP are expected to increase in the future as compared to the historical period irrespective of the climate models (GCM) or scenarios (RCP) used, as well as increased uncertainty in the future AET and NP. The parameter variability combined with climate variabilities due to GCMs and/or RCPs would cause more increased uncertainty in the future period than it appears to cause during the historical period, and it requires further investigation. The elevated water balance components as well as increased uncertainty in the simulated AET and NP rates due to combined impacts of climate and parameter variability would pose increased risks to the management of water migrating through reclaimed mine waste. The risks of increased chemical loading to the downgradient waterbodies due to increased NP rates will require to be investigated under changing climate conditions.

Table 4: Summary of water balance components for the five illustrative covers obtained using 700 sampled parameter sets with the corresponding LAI_max values (the mean LAI_max value is shown with the corresponding standard deviation)

| Illustrative cover | Precipitation | AT | AE | AET | NP | DS | LAI_max ± SD |
|---|---|---|---|---|---|---|---|
| | (mm) | (mm) | (mm) | (mm) | (mm) | (mm) | [-] |
| A50 | 426 | 297 | 99.1 | 397 | 33.3 | -3.41 | 4.12 ± 0.33 |
| A75 | 426 | 303 | 98.6 | 402 | 28.0 | -3.47 | 4.25 ± 0.30 |
| A100 | 426 | 305 | 99.8 | 405 | 25.0 | -3.53 | 4.27 ± 0.29 |
| A125 | 426 | 310 | 101 | 411 | 20.0 | -4.69 | 4.37 ± 0.27 |
| A150 | 426 | 316 | 101 | 416 | 15.0 | -5.46 | 4.50 ± 0.23 |

**3.6.2 Impact of Ks of LOS material on the simulated AET and NP**

Huang et al. (2017) show that simulated values of AET and NP are most sensitive to the distribution of the Ks of the LOS. To re-evaluate this finding in the present study, a specified range of LOS Ks values (i.e., 0.0005, 0.0023, 0.0079, 0.0270, and 0.1916 m/day) was used to simulate AET and NP over the 60-year time period for the A100 illustrative cover. A constant LAI value of 4.0 was used along with the parameter variability for the other hydraulic parameters. The results presented in Fig. 13 highlight that the mean annual AET decreases and the mean annual NP increases as Ks increases. In addition, the range of AET and NP is smallest for the lowest values of Ks. A value of Ks lower than 0.005 m/d would represent a restriction to

gravity drainage on the order of 5 mm/day, similar to a maximum rate of daily potential evaporation. As a consequence, these results are not surprising because they highlight a shift in the mechanism for water storage within the covers, specifically from being dominated by the water retention properties of the cover (i.e., high values of Ks) to 'perching' of infiltrating waters on the LOS surface (i.e., low values of Ks).

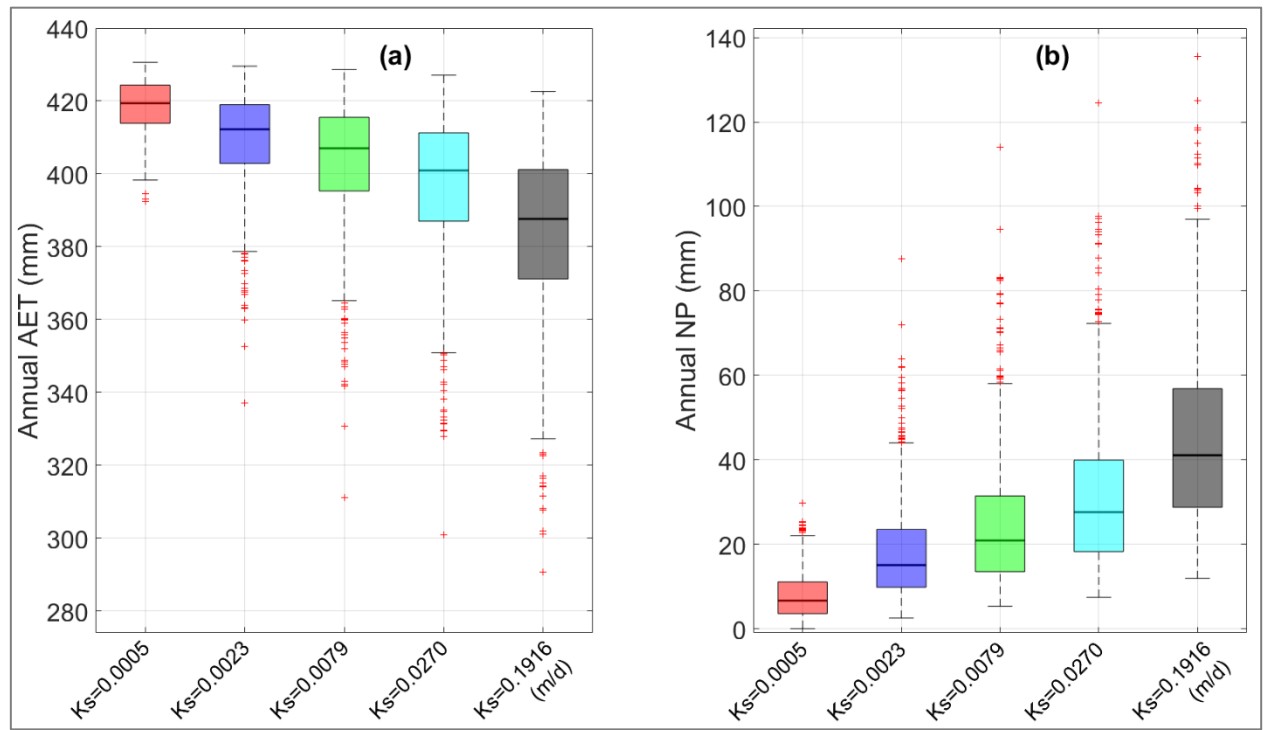

Figure 13: Box plots showing the distribution in variability in the simulated water balance components of (a) AET and (b) NP with variation in the soil hydraulic parameters, for instance, Ks of the LOS overburden. Variability of each water balance component is represented for the five illustrative covers. Description of the box plots is the same as for Fig. 11.

## 4 Conclusions

Long-term cover performance is commonly evaluated without quantification of the uncertainties which may originate from various sources including: climate, soil hydraulic parameters, vegetation index (i.e. leaf area index) etc. The use of a single set of model parameters in the design of reclamation covers precludes our ability to quantify the potential impact of uncertainties in parameters or climate and consequently makes it impossible to quantify the associated risks in performance. While our previous study (Alam et al. 2018a) investigated the impacts of climate in changing climatic conditions on the long-term cover performance, this study investigates the sources of uncertainty associated with the evaluation of long-term cover performances. This study considers a unique way to characterize the spatial and temporal uncertainty in the total parameter

uncertainty utilizing field monitoring data from 13 treatment covers (replicated in triplicate and monitored in four consecutive years). The field monitoring data includes water content, soil temperature, and soil suction values recorded at various depths at each of the treatment covers. There are few instances in the literature where such a large data record is available to quantify uncertainty. It has not been attempted previously in the context of oil sands reclamation covers. While the use of Hydrus-1D

inverse modelling tool to optimize soil hydraulic parameters (both VG and saturated hydraulic conductivity) is not new, the use of this tool to develop probability distributions for optimized parameter sets from multiple sites and years is novel, particularly when one of the parameters (Ks) can be directly compared to field measured distributions. Since, the risks associated with a cover design would be based on the probability distributions of water balance components, it seems reasonable to use a computationally efficient sampling method (PLHS in this case) to obtain all possible probability

distributions of the optimized parameters, which was motivation for the current study.

Inverse modelling (IM) in HYDRUS-1D was used to optimize five soil hydraulic parameters and produced 155 sets of soil hydraulic parameters for 13 treatment covers, replicated in triplicate, over four monitoring years. Progressive Latin Hypercube Sampling (PLHS) was used to sample parameters from the distributions of each of the five hydraulic parameters obtained from the IM approach. The randomly sampled parameters (i.e., 700 realizations) were then used in the simulations of long-term

water balance for five illustrative covers. The results from the simulations were used to highlight the coupling that occurs between the parameter variability and the LAI_max values as well as the combined impact of these variabilities on the predicted distributions of AET and NP for the five illustrative covers. Overall, the PLHS method outperforms the widely used MC sampling technique in generating the distributions of five hydraulic parameters by requiring 10 times less sample size in order to achieve similar levels of water balance components for the illustrative covers (Fig. A2 in Appendix A).

The study revealed that the peat coversoil reclamation material had the highest variability in the WRC while the LOS substrate had the greatest variability in Ks. In this study peat was combined with LFH, which might be the reason for the highest variability in WRC among the materials for three layers of the treatment covers. The results of the long-term simulated water balance highlighted how variability in climate, sampled soil hydraulic parameters, and LAI influenced the variability in both AET and NP. The distributions of the long-term simulated AET showed that climate variability exerted more impact on the

variability in annual AET. In the case of NP, the variability derived from climate and parameter variability were similar and much greater than that from LAI variability. The variability in the simulated AET values decreased with increasing cover thickness for parameter variability, while the variability in the simulated NP values decreased with increasing cover thickness for both parameter and climate variability. Median annual AET values resulting from variable parameter sets and climate data seemed to remain approximately equal for the thinner illustrative covers, while median annual NP values remained

approximately equal for the thicker covers. The PLHS and MC sampling methods produced a broader range of simulated AET and NP compared to the discrete sampling approach.

Overall, the results of this study help to highlight a wide range of cover performance risks that can occur when parameter variability is combined with climate, LAI, and cover thickness variability. The characterization of the optimized hydraulic

parameters as variable, along with the evaluation of the maximum sustainable LAI, improve our ability to characterize the uncertainty associated with the long-term simulation of cover performance beyond what is possible using a single optimized parameter set and presumed value of LAI. This study also enables an examination of how varying cover thickness changes not only cover performance (in terms of AET and NP) but also the variability in cover performance due to both climate and

parameter variability. As cover thickness increases, the annual AET increases and the annual NP decreases, as expected; however, the range in these simulated water balance components also decreases predominantly due to parameter variability. A similar insight is also available for the specific case of how variability in the Ks of the LOS affects the magnitude and range of AET and NP. In this case, the shift in cover performance combines the impact of cover thickness (i.e., water storage capacity) and the impact of restricting water flow.

Design of reclamation covers are typically based on the long-term simulations of AET and NP using a single parameter set that excludes the incorporation of parameter variability in simulating NP rates. This approach is likely to underestimate the possible ranges of NP rates. The elevated NP rates that develop when parameter variability is incorporated is an important finding which will need to be considered by industry in developing their closure designs. The consequences could be elevated volumes of water yield from the reclamation covers to the adjacent surface water bodies as well as associated increases in rates

of chemical loading from the underlying mine waste. Given the role that climate change is expected to play in future water balances of reclamation covers and the similar magnitude of impact played by parameter variability in simulating NP, integration of both climate change impacts and parameter variability across the landscapes needs to be adopted in the mine reclamation cover design in future.

In the IM modelling and long-term simulation of water balance components, the rooting depth assumption did not allow direct

water uptake from the LOS substrate. A key limitation of this study was the sole use of historical climate data without any consideration of future climate change and variability. Further research needs to examine the impact of more climate variability due to climate change (based on both Global Climate Models and Regional Climate Models) combined with these sources of variability considered here on long-term cover performance over the next few centuries.

*Data availability.* Climate and soil monitoring data for the treatment covers at Aurora North Mine site are available on the

Syncrude Watershed Research Database (https://syncrude.emline.ca/) per the outlined data policy. For the simulated water balance components (AET and NP), please contact the corresponding author.

*Competing Interests.* There are no competing interests.

*Acknowledgements.* The work was financed by the Natural Sciences and Engineering Research Council of Canada (NSERC) and Syncrude Canada Ltd. (File No. 428588-11). Special thanks are given to Amy Heidman of O'Kane Consultants Inc. for

providing uninterrupted access to the Syncrude watershed research database. We appreciate Razi Sheikholeslami and Saman Razavi for sharing the PLHS Toolbox with us. We thank Stephanie Villeneuve for designing Fig. 1a and Larisa Doucette for sharing Fig. 1b.

**Appendix A**

**Latin Hypercube Sampling (LHS) and Progressive Latin Hypercube Sampling (PLHS)**

LHS is based on the concept of a "Latin square", which forms an n-by-n matrix filled with n different objects (i.e., parameter values in this study) such that each parameter value occurs exactly once in each row and exactly once in each column. Briefly, a unit hypercube [0,1] in a multi-dimensional space (dimension is equal to the number of parameters in this study) is divided into n intervals (n is the sample size) with an equal length of 1/n. This division generates n equally probable intervals for each dimension. Sample points are then selected from each of the equally probable intervals such that the distribution of the sample points follows a uniform distribution and the sample represents a Latin hypercube. This sampling strategy ensures that sampling is representative of each equally probable interval in the total sample size. In the case of PLHS, the total sample is sliced into s slices, where the sample size of each slice is m (m = n/s). Sample points are selected for each slice that follows a uniform distribution, and the sample points from previous slices (each of which is Latin hypercube) are added sequentially such that the resulting n-point sample is a Latin hypercube. The employment of PLHS as a sampling technique ensures that the sample size from each slice is not discarded even if it fails to be an appropriate sample size. Finally, the uniformly distributed samples are transformed to the desired distributions (e.g., normal, log-normal) by the associated transformation functions. For more details, interested readers are referred to the development of PLHS by Sheikholeslami and Razavi (2017).

**Grouping of soil materials based on optimized parameters**

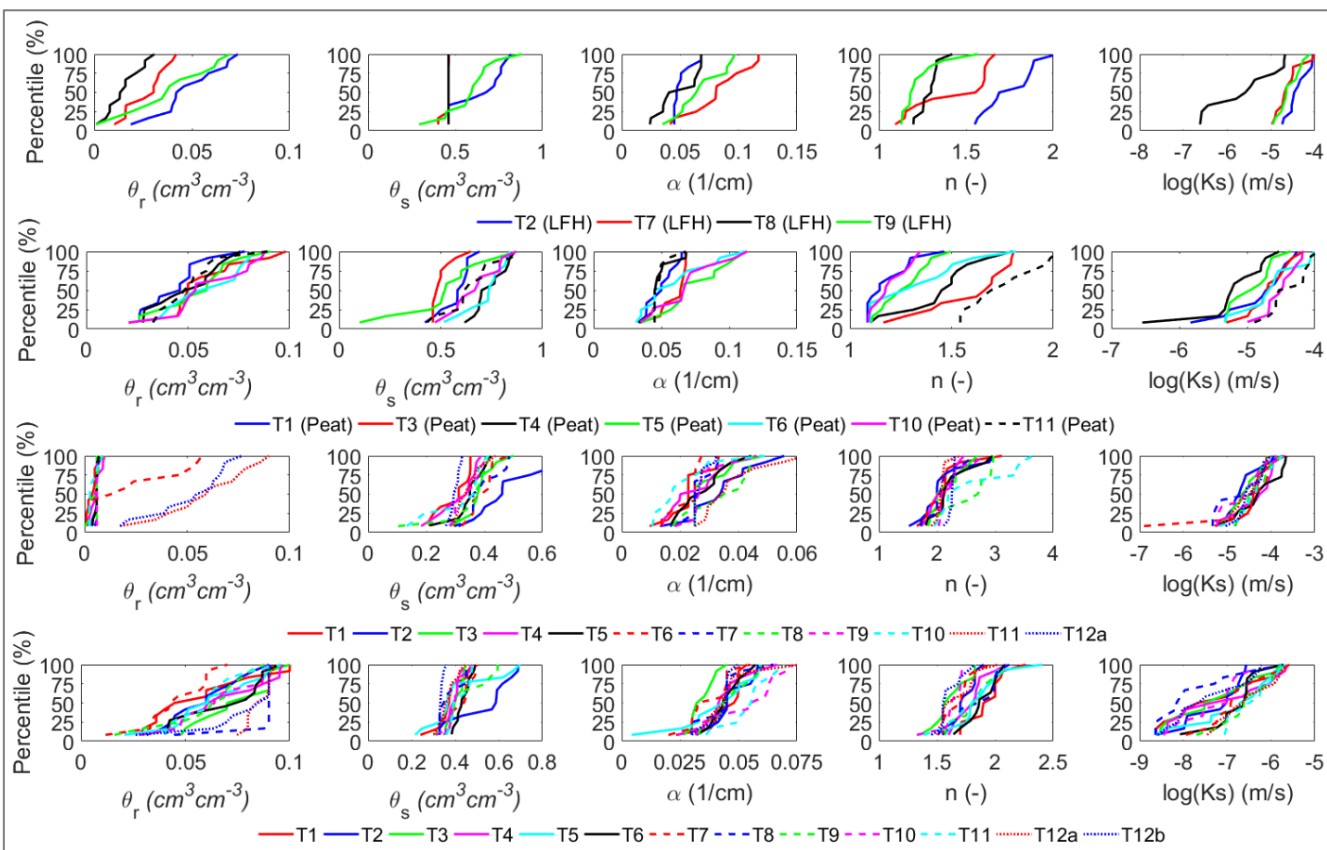

Figure A1: Frequency distributions of the optimized parameters for LFH (top panel), peat (second panel) subsoil (third panel), and LOS (bottom panel) materials

**Selection of an appropriate sample size for PLHS**

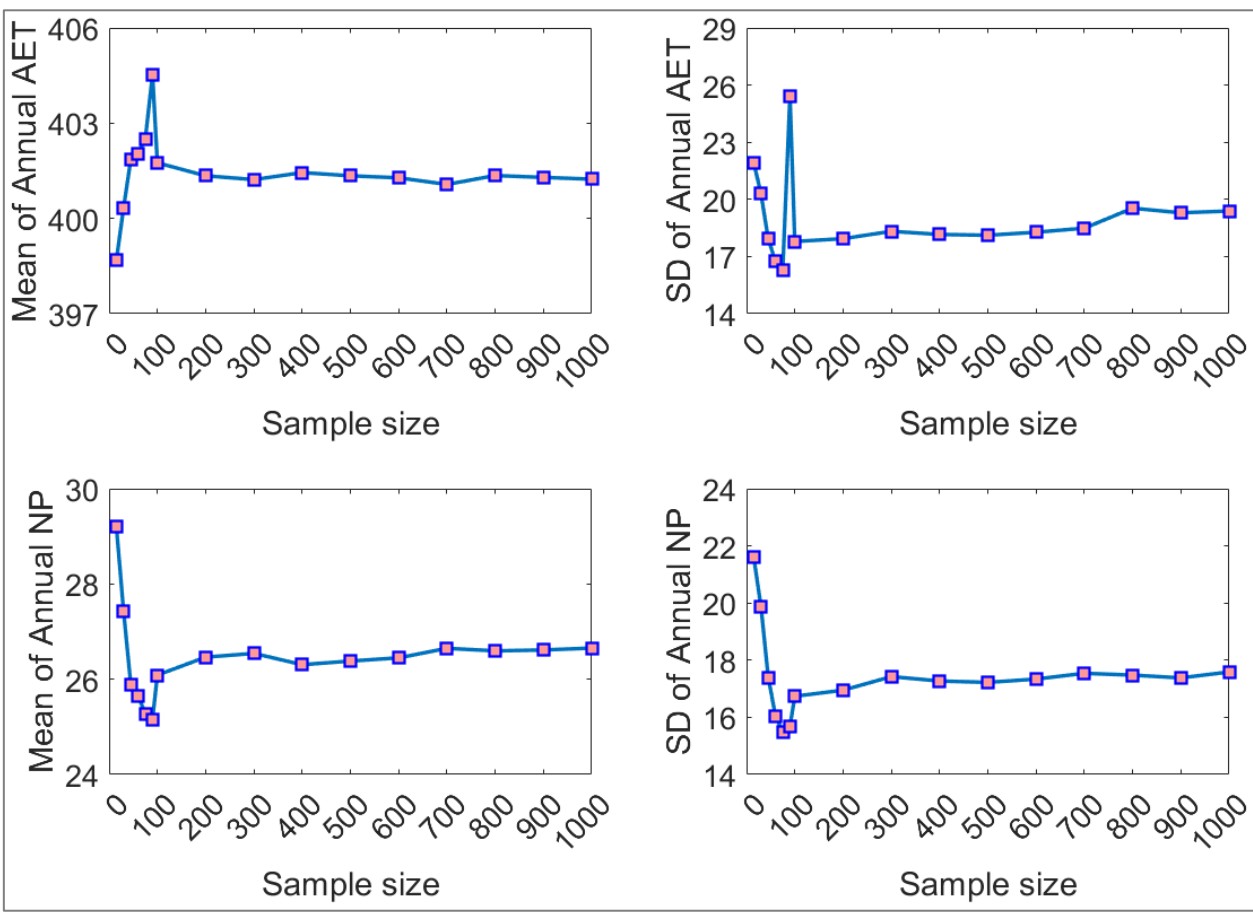

Figure A2: Mean and SD values of the annual AET and NP corresponding to each sample size

**Selection of an appropriate sample size for MC**

a.

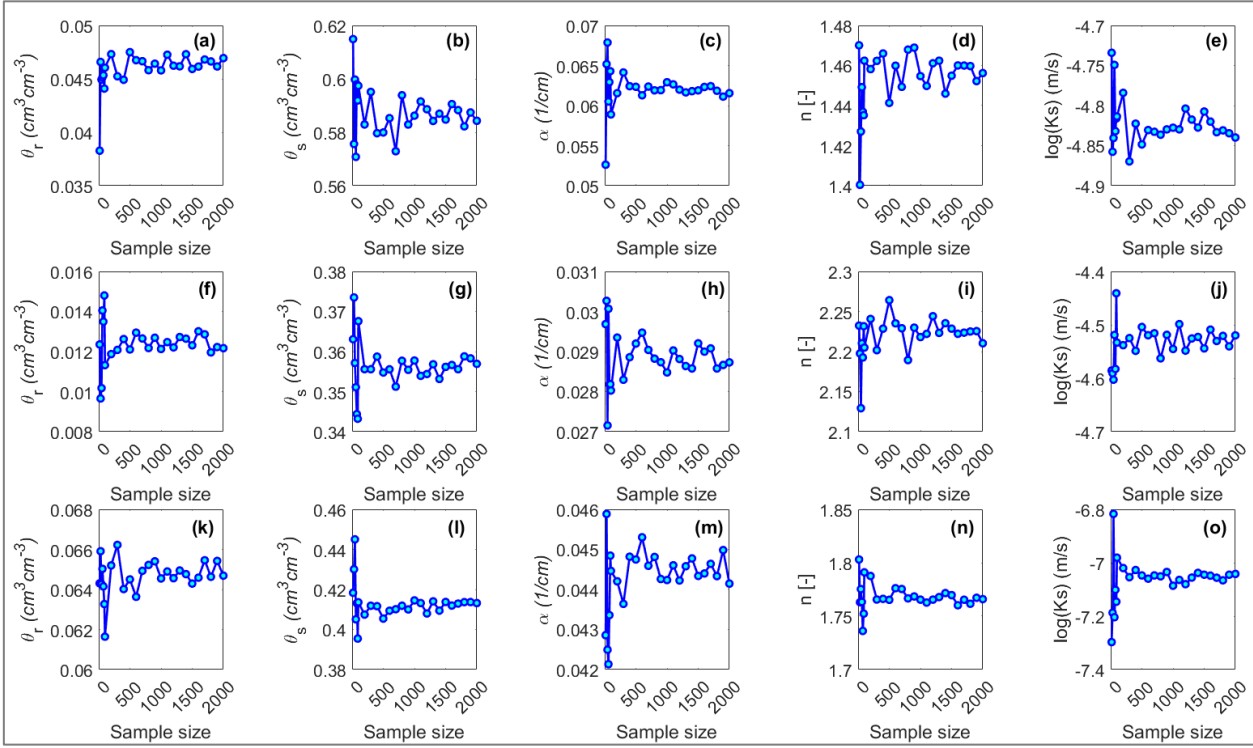

b.

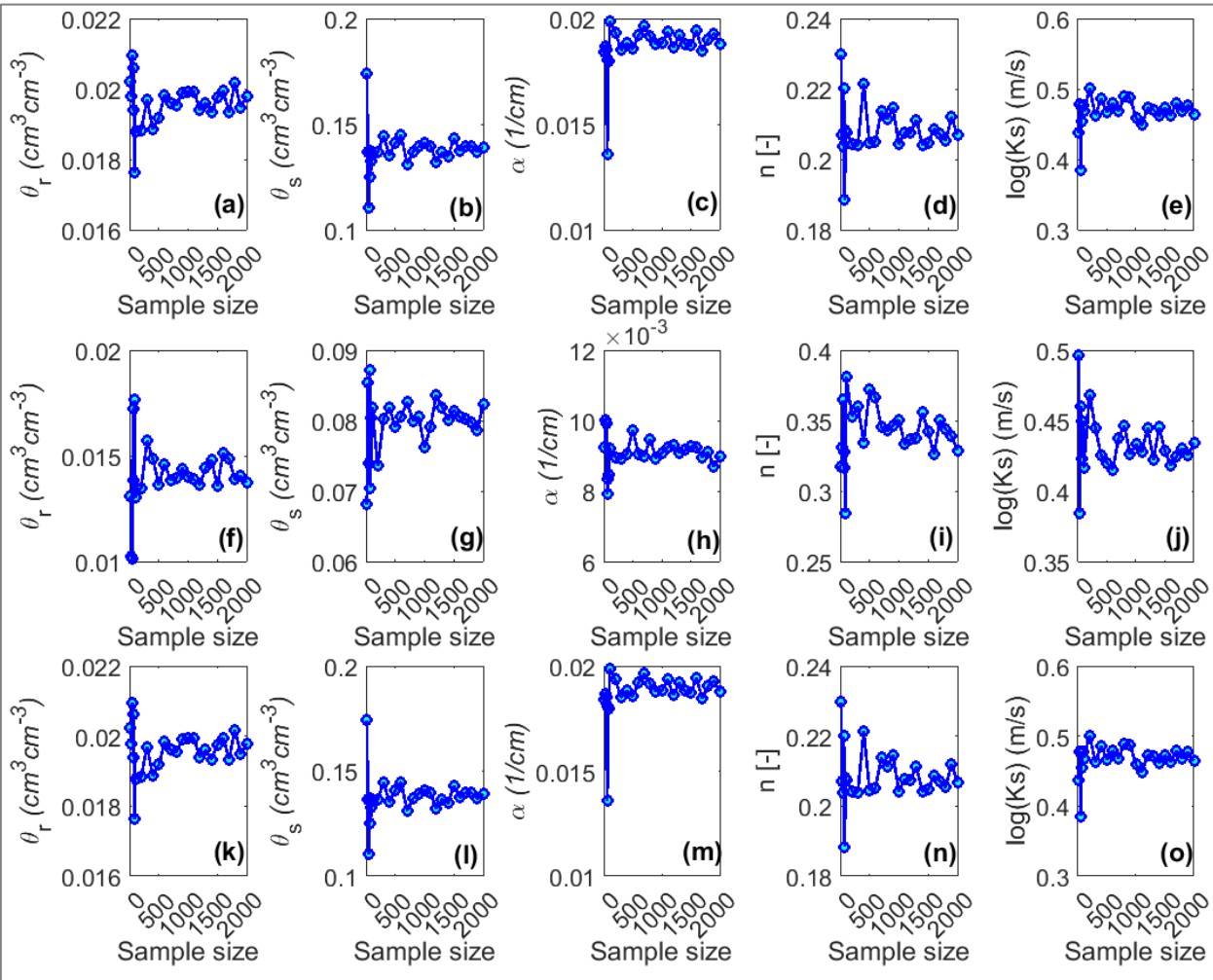

Figure A3: (a) Mean and (b) SD of the sampled parameter values corresponding to each sample size from the MC sampling method. Results are shown for peat (top row), subsoil (middle row), and LOS (bottom row) in both (a) and (b).

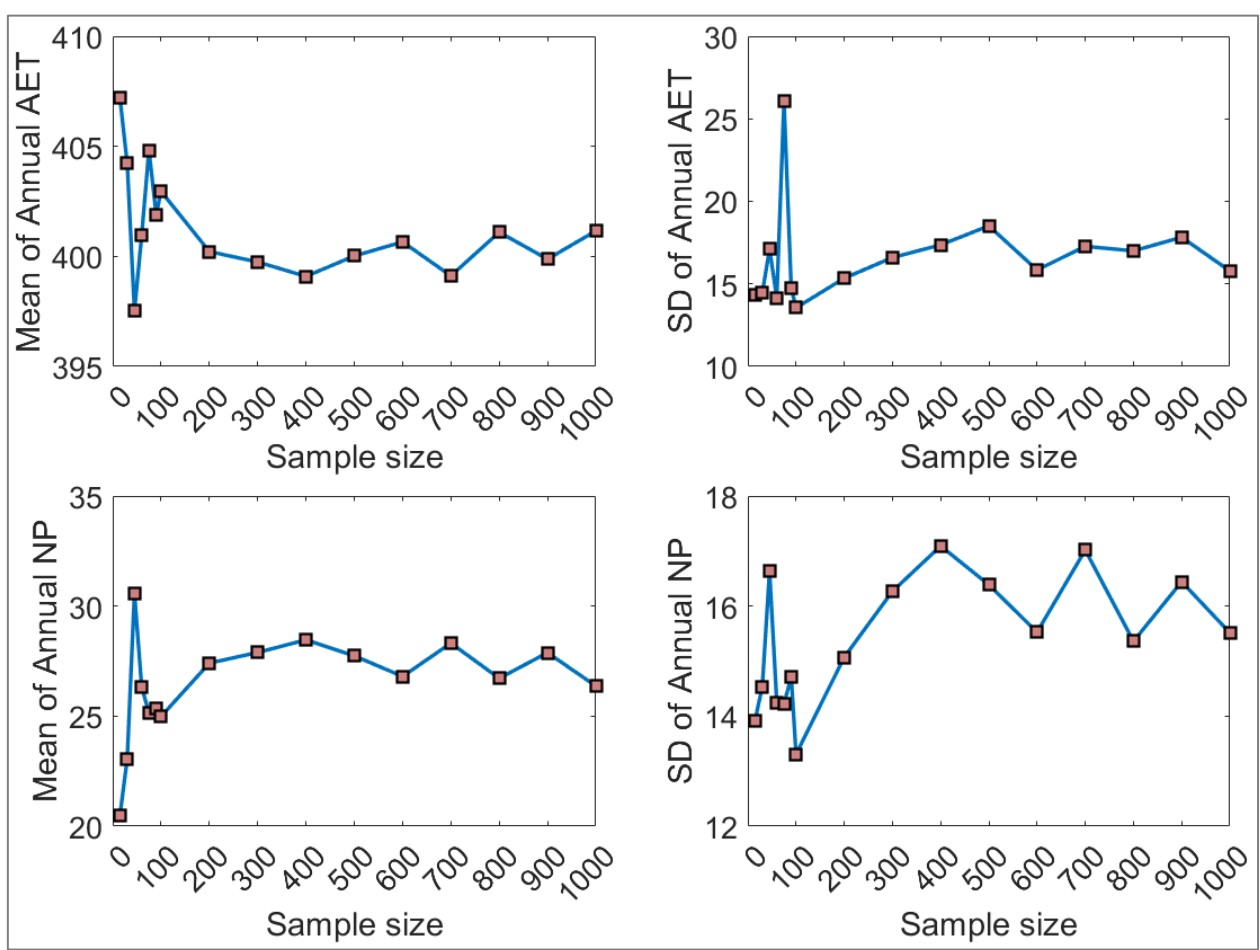

Figure A4: Mean and SD values of the annual AET and NP corresponding to each sample size for MC sampling method.

**Partition of water balance components with LAI**

This section evaluated how AET might be portioned into AT and AE as shown in Fig. A5. The results showed that almost 75%
of AET was used as AT and approximately 25% was used as AE for the five illustrative covers. However, the share of AE vs.
AT was higher at lower LAI; the share of AT monotonically increased while the share of AE monotonically decreased once
the LAI was higher than 1.5.

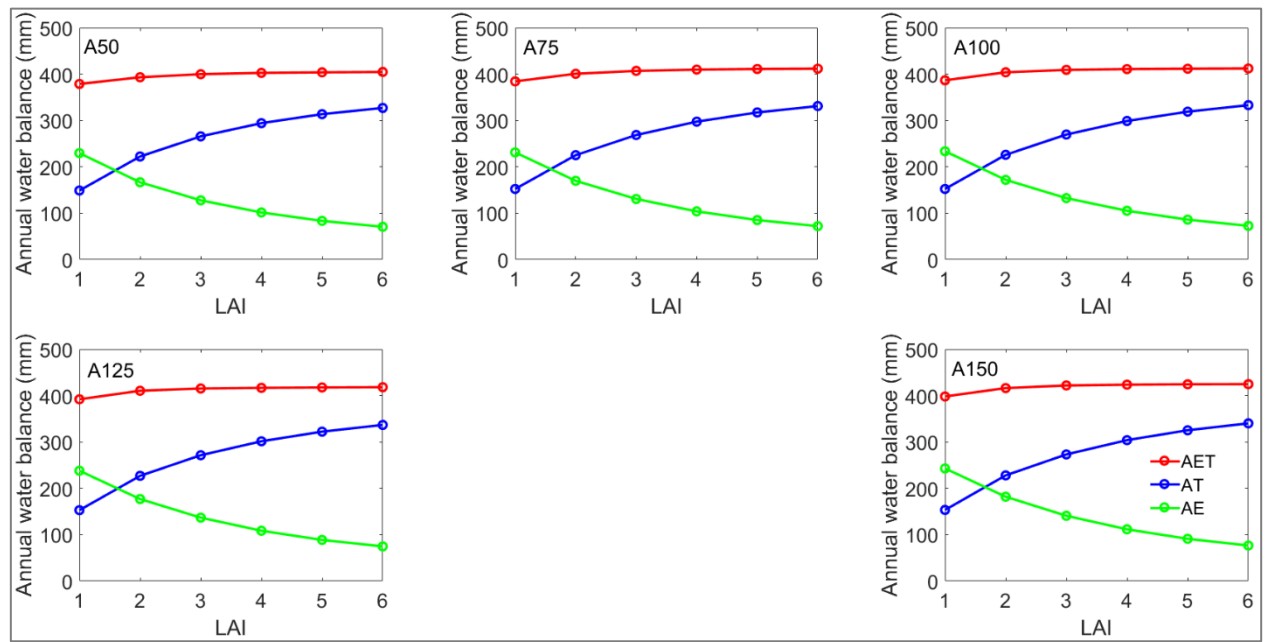

Figure A5: Partition of AET into AT and AE with the variation of LAI for the five illustrative covers

**Uncertainty in the simulated AET and NP due to sampling methods**

Figure A6 compares the distributions of the simulated water balance components (i.e., annual AET and NP) of A100 illustrative
5    cover obtained from PLHS, discrete, and MC sampling methods based on a constant LAI of 4.0. Overall, the PLHS and MC
methods show a wider inter-quartile range compared to the discrete approach in the case of AET and NP. In addition, PLHS
and MC result in slightly higher annual AET and slightly lower NP than the discrete method; however, PLHS and MC sampling
methods seem to capture similar variability as well as approximately equal median water balance components. However, the
computational effort required for MC sampling and simulation is approximately 10 times greater than for the PLHS method.

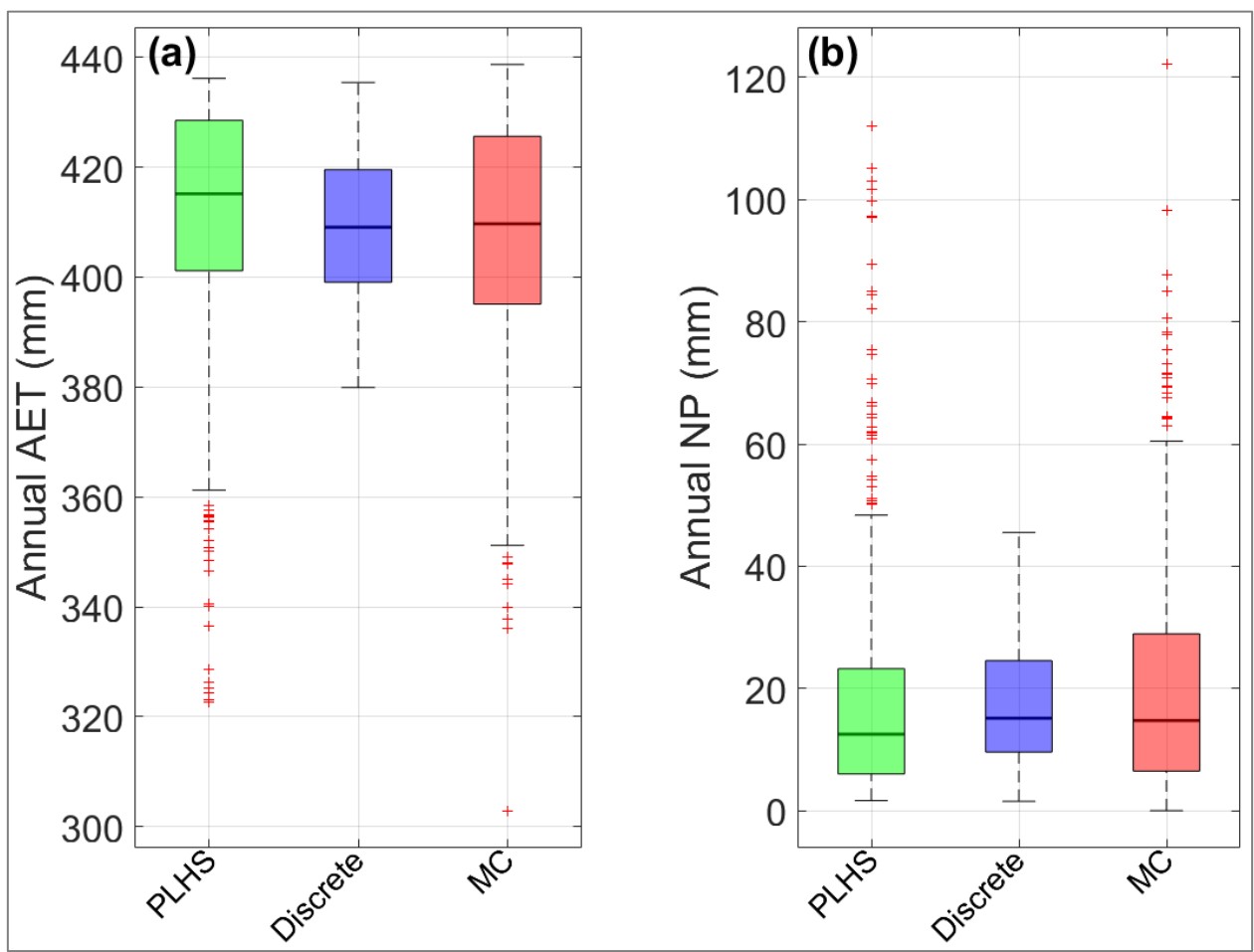

Figure A6: Box plots showing the distributions of (a) annual AET and (b) annual NP obtained from the simulated water balances for A100 illustrative cover with 700 parameter sets for PLHS (green boxes), 135 parameter sets for discrete sampling (blue boxes), and 1000 parameter sets for MC (red boxes) over a 60-year climate cycle. Description of the box plots is the same as for Fig. 11

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
