# Peer review of "Characterizing Uncertainty in the Hydraulic Parameters of Oil Sands Mine Reclamation Covers and its Influence on Water Balance Predictions"

_Hydrology and Earth System Sciences, 2019_

## Referee Comment (RC1) · Anonymous Referee #1 · 17 Jul 2019

General Comments

This paper evaluates soil hydraulic property uncertainty based upon multiple realizations of inverse modelling of soil reclamation covers. They use an innovative technique, Progressive Latin Hypercube Sampling (PLHS), to generate a large data set (statistical distribution) of optimized parameters. The most consequential finding showed that climate variability results in greater variability than uncertainty and heterogeneity in soil hydraulic properties. The results are of interest to the academic community and potentially to industry, however there are several issues with the underlying model that

require further attention.

The entire study rests on the initial inverse modelling that identified a group of soil hydraulic properties that could then be used to generate a statistical distribution of optimized properties. However, any assessment of model performance is absent from the manuscript, and the only graphical representation of model fit is relegated to the appendix and only shows simulated versus observed r2, which is not enough for the reader to judge the performance of the model. Since identifying appropriate soil hydraulic properties is so critical to the rest of the study, model performance should feature much more prominently.

Furthermore, I would like to be convinced that the inverse optimization did, in fact, find the appropriate properties. The internal HYDRUS inverse scheme used is notorious for being unable to find the global minima for each parameter, and getting "stuck" in local minima. If using this type of inverse scheme, the appropriate methodology is to vary the initial parameter "guess" to ensure that the global minima has been identified. Was this was done? Also, the range over which the optimized algorithm was allowed to vary was extremely small, and could easily be excluding the global minima. Further explanation of the approach to inverse modelling, and justificatiojn of the narrow range for optimization is needed. Further clarity is also needed on how many parameters were simultaneously inversed, as there is a preponderance of evidence that demonstrates the inability of gradient-based inverse schemes to identify global minima when more than 6 parameters are inversed (the developer of HYDRUS has stated that he has never successfully optimized more than 5 or 6 parameters).

Besides the aforementioned technical criticisms, this manuscript lacks (in my opinion) a genuine discussion section that ties the results to an interpretation of the physical characteristics of the soil and its setting. I would have welcomed an expanded discussion of relevant physical processes on the reclamation landscape that the results pertain to. This could be done throughout the results and discussion section. Furthermore, while the authors honestly admit the limitation that no consideration was given

to climate change, it seems a small leap to say that if the current situation illustrates that climate variability is the main cause of uncertainty, a changing climate will magnify that. What are the implications for practice? For the most part, the conclusion section summarized the findings (i.e. was a summary, rather than conclusions). Some more insight into how this information can be used would be helpful to the reader. What recommendations would the authors make to industry trying to reclaim landscapes?

Specific Comments

P3L5-15 Given the relatively "light" computational load of HYDRUS 1D is the latin hypercube technique necessary?

P3L5-30 Latin hypercube discussion would be of little importance or significance to most readers, perhaps some of the methodology could be moved to an appendix?

P6L7 I've seen a number of different definitions for the initialism LFH, but never leaf, folic, and humic. Perhaps look at the definitions of Naeth?

P6L6-18 This section is pretty hard to follow, suggest rewording

P6L19 Particle size distributions for peat are not common due to its fibric nature. A brief note on what was done would be appropriate

P7L1 It does not really seem appropriate to lump the peat and LFH covers into a single group

P8L4-L32 More detail is needed here. What about the Mualem tortuosity parameter? Was this inversed or set to equal 0.5 or some other value? Did each simulation include the inversing of multiple materials? If so this could be highly problematic, obtaining parameter estimates through inverse simulation is exceptionally difficult with more than 5 or 6 parameters (as stated by Simunek), let alone 15. Furthermore, the HYDRUS inverse method of Marquardt-Levenberg is highly susceptible to getting "stuck" in local minima as opposed to finding the global minima. In order for successful inverse simulation to find the global minima with this method the initial values must be varied. If the

same parameter sets are found after this procedure then it can be assumed with reasonable confidence that the global minima has been identified. All subsequent findings require that you found appropriate soil hydraulic properties. I am not yet convinced that you did, I would need to see more results related to the performance of the inverse simulation.

Furthermore, as shown by the parameter values in Table 1 many of the parameters are extraordinarily tightly constrained (alpha, n, and Ks in particular). Even covers of the same (ostensible) material could easily vary by multiple orders of magnitude, yet only the lean oil sands are allowed to vary more than roughly 1 order of magnitude. The van Genuchten n parameter for the subsoil has remarkably little freedom to vary.

Also I could not seem to find any mention of root water uptake parameters, or the parameters associated with partitioning E and T from PET, or the interception constant.

P9L19 In the Discussion, I would like to have your assessment of the implications of this choice of root distribution, since it is idealized and probably incorrect.

P10L2 Unit gradient meaning free drainage?

P10L25. Are these (optimized soil parameters) the V-G parameters?

P11L11 I'm not clear on what you mean by discrete parameter distributions not being representative of the range of distributions

P12L2 Just to be clear, this is the order of the soil profile? Peat/LOS/Subsoil?

P12L6 There is evidence to suggest that some early covers generate a lot of snowmelt runoff, rather than infiltrating into the soil (e.g. Ketcheson et al., 2016), although others that suggest mostly infiltration (Nichols et al., 2016). Are there measurements or observations at this site that support your assumption. You should probably include a statement to this effect.

P12 L7 Does this method consider sublimation?

P13 L9 The performance of your model should not be hidden in an appendix, and it is not acceptable to just show the r2, if only a single model metric were chosen, show the Kling-Gupta Efficiency. Otherwise, the RMSE, should be shown as well as perhaps graphs over time of the simulated and observed water contents.

P13L13 This seems poor justification for grouping the parameter sets together.

P14L1 Why would you be interested in including temporal variability into these parameter sets? By then applying these parameter sets to long-term simulations you would be artificially accentuating the variability in net percolation and ETa.

P14L5 I don't think that you have shown enough supporting evidence to make that assertion. Did you perform this same procedure for alpha, n, theta r, and theta s? P17L2 Does this approach assume that there is no correlation between parameters?

P21L8 Are you surprised that the LAI max occupies such a narrow range for such different soil covers? Why is the range so small? Please add a comment on this, as 4.12 to 4.50 are basically the same, as far as model precision goes. Doesn't this suggests that you can get a reasonable forest growing on basically any cover? The differences in Figure 10 are so slight it seems like the same graph copied 5 times.

P22L9 and Fig11 Are there statistically significant differences?

P27L10 I'm not really sure I understand the value of this whole objective and section. I would remove this from the manuscript.

P29L9 This seems to be a foregone conclusion since you have already admitted to lumping to very different materials into this category.

P29L19-20 "the results of this study help to highlight a wide range of cover performance risks that can occur when parameter variability is combined with climate, LAI, and cover thickness variability", this is really the first and only mention that the reader gets of cover performance risks. I am interested to know more, but it needs to come before that statement!

P29L25 AET really didn't change that much in any scenario. The data seems to conflict with the interpretation.

Fig. 3 I could be mistaken but it seems that the hydraulic conductivities are outside the maximum and minimum values that are seen in Table 1. The fact that some parameters are logged and others not makes it difficult to compare. The alpha values in the figure are in different units than what is in Table 1 (not the only instance of alpha values being in inconsistent units). The saturated water content of the subsoil seems to in some cases be very low. What is the cause of the inverse scheme identifying a 0.1 or 0.2 water content? That is almost certainly incorrect.

Fig. A2 While it seems reasonable that the subsoil and lean oil sands are grouped together. It does not seem appropriate that the peat and LFH soils are grouped. Their parameters (particularly alpha and n) differ substantially.

Formatting Comments

Fig. 1 Font in legend is unclear - small and resolution too poor. Figure should be improved for readability

Fig. 2 Should be improved for readability, very difficult to understand

Fig. 9 Could probably be moved to an appendix

Fig 11 could be replaced with another line for LAI+/- SD in Table 3.

Fig 12&13 could be put into a single Table (there are a lot of Figures).

Fig 15 could be summarized in one or two sentences in the text.
* * *

---

## Referee Comment (RC2) · Claire Cote (Referee) · 29 Aug 2019

General Comments The objectives of this manuscript are not clear, are focused on minor modelling considerations and do not contribute valuable findings to the field of waste cover design. This is unfortunate as it forms part of a broader program of work that would bring interesting findings worth publishing. For mine reclamation covers, key objectives of the design should be that (1) it will lead to successful ecosystem restoration ( eg. plants are growing successfully) (2) the quality of the seepage does not lead to impacts on the receiving environment and (3) it addresses any other objective identified as part of design and mine closure planning. The manuscript does not clearly state how it contributes to the understanding of how the selected cover designs will meet these objectives. Among other things, models are used to provide a simplified mathematical representation of reality so that various scenarios can be analysed. If we change the parameters in the mathematical representation, the results from the model will be different. It does not meant the reality itself has changed, so I struggle to understand what the study is trying to prove. It seems like a circular proposition. If we modify the van Genuchten parameters then of course the predicted water balance will be different. The issue is not the value of the parameters but our ability to understand how changes to the water balance can prevent the design from meeting the objectives stated above. For instance, a design is developed specifically to minimise seepage but a modification in the surrounding environment leads to increased seepage and risk to the receiving environment. Statistical analysis can assist with quantifying the probability of that scenario and this is what we should be focusing on. If the scenario is likely then there is a requirement to develop mitigation options, such as installation of additional monitoring equipment to capture changes and mitigate them. The values of the hydraulic parameters in the mathematical representation are not that relevant to the mitigation of the issue. With advances in mathematical software, particularly Mathematica, MatLab or R, there is less and less of a need to develop specific mathematical approaches for solving soil science or environmental science problems. With the Wolfram Language now in the public domain, it is even less justifiable. I have never heard of the Latin Hypercube Sampling, but looking at the description in the manuscript, all I would need to do in Mathematica is to Map the Range function unto the parameters. It seems very straight forward and certainly does not justify a publication. Hydrus 1D could probably be re-coded in Mathematica very quickly and the predictions would be much faster and much more stable. It does not seem that there has been much evolution in the field of vadose zone hydrological modelling in the last 20 years. There would be value in re-assessing these models in light of advances in mathematical software. In terms of developing a more robust approach to test the performance of cover design,

I would suggest leaching experiments on large columns, which can reproduce leaching results much more quickly than what can be obtained in the field. It is a more reliable method to investigate the impact of variations in the water balance. The greatest long term risk to the viability of soil covers and associated ecosystems is climate change with associated variations in temperature and precipitation. Column tests can provide an indication of expected changes in soil water behaviour under extreme precipitation or temperature regimes. This would be a much more robust contribution than mathematical manipulations of soil parameters. Another key gap seems to be the assessment of snow pack height on cover behaviour. This could also be investigated with column testing. Assessment against HESS criteria The paper is well-written, well-referenced and well-structured. Figures, tables and cross-references are fine. The key issue relates to the value of the contribution to the hydrological community. The objectives and research questions are ill defined. The paper does not present novel concepts, ideas or tools. Clear conclusion are provided but they do not contribute much to the assessment of the performance of cover designs. The description of experiments and calculations are complete and precise. The authors give proper credit to related work. The work described in the paper is part of a broader project that is very interesting. The title clearly reflects the contents of the paper. The abstract provides a concise and complete summary.The language is fluent and precise. The number and quality of references appropriate There is no need to provide details about the Latin Hypercube Sampling methodology, it is a routine calculation.

---

## Author Comment (AC1) · 31 Aug 2019

In the attached file (ZIP file), we have fully addressed RC1's concerns in the highlighted manuscript (HESS Manuscript-2019-154-Highlighted.pdf) and provided a point-by-point response in another file (Response to RC1.pdf).

Please also note the supplement to this comment:
https://www.hydrol-earth-syst-sci-discuss.net/hess-2019-154/hess-2019-154-AC1-supplement.zip

---

## Author Comment (AC2) · 31 Aug 2019

In the supplement (ZIP file), we have fully addressed RC2 Dr. Claire Cote's concerns in the highlighted manuscript (HESS Manuscript-2019-154-Highlighted.pdf) and provided a point-by-point response in another file (Response to RC2.pdf).

Please also note the supplement to this comment:
https://www.hydrol-earth-syst-sci-discuss.net/hess-2019-154/hess-2019-154-AC2-supplement.zip

---

## Author Response (AR1)

*Responses to the Anonymous Referee #1*
*The detailed review by Anonymous Referee 1 and his/her positive feedback on our paper are greatly appreciated. Below we attach our brief response to the issues raised by the reviewer in expectation of a successful discussion.*

General Comments

This paper evaluates soil hydraulic property uncertainty based upon multiple realizations of inverse modelling of soil reclamation covers. They use an innovative technique, Progressive Latin Hypercube Sampling (PLHS), to generate a large data set (statistical distribution) of optimized parameters. The most consequential finding showed that climate variability results in greater variability than uncertainty and heterogeneity in soil hydraulic properties. The results are of interest to the academic community and potentially to industry, however there are several issues with the underlying model that require further attention. The entire study rests on the initial inverse modelling that identified a group of soil hydraulic properties that could then be used to generate a statistical distribution of optimized properties. However, any assessment of model performance is absent from the manuscript, and the only graphical representation of model fit is relegated to the appendix and only shows simulated versus observed r2, which is not enough for the reader to judge the performance of the model. Since identifying appropriate soil hydraulic properties is so critical to the rest of the study, model performance should feature much more prominently.

*Response: The performance of the inverse modelling technique of Hydrus-1D model was first evaluated by comparing the measured and simulated water contents at various depths within each of 13 treatment covers. The coefficient of determination ($R^2$) and root-mean-square errors (RMSE) between the measured and simulated water contents are shown in Table 2, while the comparison between the measured and simulated water contents at various depths within each of 13 treatment covers in a typical year during 2013-2016 is shown in Fig. 3. For the treatment covers, the $R^2$ values are mostly above 0.8, and RMSE values are mostly less than 1 mm/day, except for a few treatment covers. The performance criteria as well the graphical comparison between the measured and simulated water contents at various depths within the treatment covers show that the models perform reasonably well given diverse soil conditions, number of treatment covers, and number of parameters to be optimized. (P18L16-24)*

*Table 2: Performance statistics ($R^2$ and RMSE) of inverse modelling for each of 13 treatments covers at the Aurora North Mine site (P19)*

| Treatment cover # | $R^2$ | RMSE (mm/day) |
|---|---|---|
| 1 | 0.89 | 0.66 |
| 2 | 0.82 | 0.57 |
| 3 | 0.73 | 0.40 |
| 4 | 0.81 | 0.79 |
| 5 | 0.62 | 1.00 |
| 6 | 0.86 | 1.07 |
| 7 | 0.79 | 0.34 |
| 8 | 0.82 | 0.39 |
| 9 | 0.51 | 1.06 |
| 10 | 0.84 | 0.72 |
| 11 | 0.84 | 0.71 |
| 12a | 0.81 | 0.28 |
| 12b | 0.90 | 0.29 |

*A typical comparison of soil water contents at various depths within each of 13 treatment covers is shown in Figure 1. The water content values are compared between the measured and simulated data only for the days when the treatment covers were unfrozen (i.e. temperature greater than $0\,^oC$).*

[Figure]

*Figure 1: Comparison between the measured and simulated water contents at different depths within each of 13 treatment covers for the days when temperature is greater than 0 °C. Typical depths at which the water content measurements are recorded vary from 5 to 200 cm within the treatment covers (P20L1-5)*

*The performance criteria as well the graphical comparison between the measured and simulated water contents at various depths within the treatment covers show that the models perform reasonably well given diverse soil conditions, number of treatment covers, and number of parameters to be optimized. (P18-20)*

Furthermore, I would like to be convinced that the inverse optimization did, in fact, find the appropriate properties. The internal HYDRUS inverse scheme used is notorious for being unable to find the global minima for each parameter, and getting "stuck" in local minima. If using this type of inverse scheme, the appropriate methodology is to vary the initial parameter "guess" to ensure that the global minima has been identified. Was this was done? Also, the range over which the optimized algorithm was allowed to vary was extremely small, and could easily be excluding the global minima. Further explanation of the approach to inverse modelling, and justificatiojn of the narrow range for optimization is needed. Further clarity is also needed on how many parameters were simultaneously inversed, as there is a preponderance of evidence that demonstrates the inability of gradient-based inverse schemes to identify global minima when more than 6 parameters are inversed (the developer of HYDRUS has stated that he has never successfully optimized more than 5 or 6 parameters).

*Response: In this study, the ranges of initial parameter values were estimated from the measured PSDs and bulk density using Arya-Paris model (Arya et al. 1999). The water retention curves (WRC) for each PSDs from peat/LFH, subsoil, and LOS were estimated using the equations presented in the Arya-Paris model and the least-square optimization program RETC (van Genuchten et al. 1991) was used to fit the VG-Mualem equation to the estimated WRC from Arya-Paris model to estimate the VG parameters ($\theta r$, $\theta s$, $\alpha$, n). The Kozeny-Carman equation (Kozeny 1927; Carman 1938, 1956) was used to estimate Ks values from the PSDs as it is one of the most widely used and accepted methods (Huang et al. 2011a; Mathan et al. 1995). The estimation of parameters using these methods helps to constrain the initial parameter ranges in the inverse modelling. In addition to the Ayra-Paris model, the initial range of $\theta s$ can also be approximated from the measured volumetric water content (vwc) data for the covers, where the maximum water content values are observed at the depths of 5-10 cm. After setting up the initial range of parameter values based on the above methods, the inverse modelling is repeated with different initial values. Once there is no significant change in $\theta r$ and $\theta s$ parameters and objective function (i.e. sum of least squares), these parameters are assumed optimized and kept fixed in the subsequent inverse modelling for the remaining parameters. Step-by-step the least sensitive parameters are kept fixed and thereby reducing the number of parameters to be optimized by inverse modelling. Reducing the number of parameters, constraining the range of initial parameter values, and repeating the inverse modelling with initial parameter values were done as recommended by Hopmans et al. (2002). However, details of all these steps are not included in this manuscript, only referenced to Hopmans et al. (2002), for the brevity of the manuscript. It is important to note that the purpose of this manuscript was not to focus on inverse modelling techniques but rather to highlight how reasonably optimized parameter sets can resemble the distribution of the measured key parameter (i.e. Ks) and represent the parameter variability. This comparison between the optimized and measured key parameter values was assumed an indirect validation of the inverse modelling approach used in this study, which can be used for further sampling based on PLHS with certain level of confidence. We incorporated this briefly in the revised manuscript. (P13L4-23)*

Besides the aforementioned technical criticisms, this manuscript lacks (in my opinion) a genuine discussion section that ties the results to an interpretation of the physical characteristics of the soil and its setting. I would have welcomed an expanded discussion of relevant physical processes on the reclamation landscape that the results pertain to. This could be done throughout the results and

discussion section. Furthermore, while the authors honestly admit the limitation that no consideration was given to climate change, it seems a small leap to say that if the current situation illustrates that climate variability is the main cause of uncertainty, a changing climate will magnify that. What are the implications for practice? For the most part, the conclusion section summarized the findings (i.e. was a summary, rather than conclusions). Some more insight into how this information can be used would be helpful to the reader. What recommendations would the authors make to industry trying to reclaim landscapes?

*Response: We thank the reviewer for highlighting this shortcoming. We incorporated a more detailed discussion of these issues in the revised manuscript; however, we would like to state some of key discussions here as raised by the reviewer. Below are a couple of examples:*

*(i) P32L1-4: Huang et al. (2015) showed that the increases in AET are not necessarily proportional to the incremental increases in cover thickness, rather little increment is noticed in the median AET over a climate cycle once a threshold cover thickness is passed. Therefore, it is not a surprise to observe the narrow range of LAI_max values as shown in Fig. 10. That said, there is support for decreased NP rates for thicker covers as greater volumes of water can be stored and ultimately released as AET.*

*(ii) P37L5-11: The results showed that climate variability is a key source of uncertainty for the simulated AET during the historical 60-year period. That said, climate and parameter variabilities appear to cause similar levels of uncertainty in the simulated NP rates during the same historical period. Our previous studies (Alam et al. 2017b, 2018a) showed that the median AET and NP are expected to increase in the future as compared to the historical period irrespective of the climate models (GCM) or scenarios (RCP) used, as well as increased uncertainty in the future AET and NP. The parameter variability combined with climate variabilities due to GCMs and/or RCPs would cause more increased uncertainty in the future period than it appears to cause during the historical period, and it requires further investigation.*

*We provided some recommendations in the revised manuscript "Design of reclamation covers are typically based on the long-term simulations of AET and NP using a single parameter set that excludes the incorporation of parameter variability in simulating NP rates. This approach is likely to underestimate the possible ranges of NP rates. The elevated NP rates that develop when parameter variability is incorporated is an important finding which will need to be considered by industry in developing their closure designs. The consequences could be elevated volumes of water yield from the reclamation covers to the adjacent surface water bodies as well as associated increases in rates of chemical loading from the underlying mine waste. Given the role that climate change is expected to play in future water balances of reclamation covers and the similar magnitude of impact played by parameter variability in simulating NP, integration of both climate change impacts and parameter variability across the landscapes needs to be adopted in the mine reclamation cover design in future." (P42L12-20)*

Specific Comments
P3L5-15 Given the relatively "light" computational load of HYDRUS 1D is the latin hypercube technique necessary?

*Response: While Hydrus-1D can be used to optimize parameters with reasonable computational cost, our goal was simply to use Hydrus-1D to optimize a set of parameters for each cover with each year's monitoring data. Thus, we obtained 155 sets of parameters which include 13 treatment covers, replicated in triplicate and monitored in four consecutive years. Since these parameters form a distributions of parameters representative of the measured parameter distributions (at least for Ks), we decided to use a standard sampling technique (e.g. PLHS) to do the rest with regards to generating multiple sets of*

*parameters. Comparison between the multiple sampling from Hydrus-1D and from PLHS could be an extended study of the current in terms of both performance and computational cost. (P16L18-25)*

P3L5-30 Latin hypercube discussion would be of little importance or significance to most readers, perhaps some of the methodology could be moved to an appendix?

*Response: We moved the redundant part of LHS and PLHS methods to the appendix in our revised manuscript. (P3L9-15)*
*Appendix A now includes the following:*
*"Latin Hypercube Sampling (LHS) and Progressive Latin Hypercube Sampling (PLHS)*
*LHS is based on the concept of a "Latin square", which forms an n-by-n matrix filled with n different objects (i.e., parameter values in this study) such that each parameter value occurs exactly once in each row and exactly once in each column.*
*Briefly, a unit hypercube [0,1] in a multi-dimensional space (dimension is equal to the number of parameters in this study) is divided into n intervals (n is the sample size) with an equal length of 1/n. This division generates n equally probable intervals for each dimension. Sample points are then selected from each of the equally probable intervals such that the distribution of the sample points follows a uniform distribution and the sample represents a Latin hypercube. This sampling strategy ensures that sampling is representative of each equally probable interval in the total sample size. In the case of PLHS, the total sample is sliced into s slices, where the sample size of each slice is m (m = n/s). Sample points are selected for each slice that follows a uniform distribution, and the sample points from previous slices (each of which is Latin hypercube) are added sequentially such that the resulting n-point sample is a Latin hypercube. The employment of PLHS as a sampling technique ensures that the sample size from each slice is not discarded even if it fails to be an appropriate sample size. Finally, the uniformly distributed samples are transformed to the desired distributions (e.g., normal, log-normal) by the associated transformation functions. For more details, interested readers are referred to the development of PLHS by Sheikholeslami and Razavi (2017)." (P43L4-17)*

P6L7 I've seen a number of different definitions for the initialism LFH, but never leaf, folic, and humic. Perhaps look at the definitions of Naeth?

*Response: We included the definition of LFH mentioned in Naeth et al. (2013) and defined by Soil Classification Working Group (1998) in our revised manuscript. The Soil Classification Working Group in Canada defined LFH as "organic soil horizons (L, F, H) developed primarily from the accumulation of leaves, twigs and woody materials, with or without a minor component of mosses, that are normally associated with upland forest soils with imperfect drainage or drier". The L, F, and H horizons are characterized by the accumulation of original organic matter, partially decomposed organic matter, and decomposed organic matter, respectively.(P8L10-14)*

P6L6-18 This section is pretty hard to follow, suggest rewording

*Response: We have rewritten this section in the revised manuscript as follows:*
*"All the treatment covers within the ASCS were constructed in 2012 using three distinct soil layers including: coversoil, subsoil, and LOS. The coversoil was utilized in the treatment covers was either salvaged peat or LFH material. The peat was predominantly organic material with a total organic carbon of about 17% (by weight), while the general texture of the mineral component of LFH was sand (about 92% by mass). The coversoil was underlain by different selected coarse-textured subsoils salvaged from different locations (i.e. depositional environments) and depths within the mine site (Soil Classification*

*Working Group, 1998). In general, the subsoil texture is sand (92%-95% by mass). The bottom layer was constructed using LOS overburden materials that was overlain by coversoil and subsoil layers. The LOS materials consist of loamy sand to sandy loam with an oil content of 0.1% to 7.7% (NorthWind Land Resources Inc., 2013). Overall, the LOS comprises a range of different oil contents and particle sizes compared to the coversoil and subsoil materials." (P8L9-20)*

P6L19 Particle size distributions for peat are not common due to its fibric nature. A brief note on what was done would be appropriate

*Response: The particle size distribution (PSD) for the cover materials including peat was conducted using the standard analysis method of ASTM D422 by MDH Engineered Solutions in November of 2009. The ASTM D422 method is based on the assumption that the particles are spherical in shape, so the PSD for peat may not be representative. (P9L11-13)*

P7L1 It does not really seem appropriate to lump the peat and LFH covers into a single Group

*Response: According to Syncrude Canada Ltd., in the final cover design the top layer might be either peat/LFH or combination of the two, the distributions of parameters for these two materials together seem reasonable to be used in the illustrative covers for long-term simulation of water balance components. Therefore, the PLHS method was used to randomly sample from the distributions of the two materials grouped together. (P9L19-22)*

P8L4-L32 More detail is needed here. What about the Mualem tortuosity parameter? Was this inversed or set to equal 0.5 or some other value? Did each simulation include the inversing of multiple materials? If so this could be highly problematic, obtaining parameter estimates through inverse simulation is exceptionally difficult with more than 5 or 6 parameters (as stated by Simunek), let alone 15. Furthermore, the HYDRUS inverse method of Marquardt-Levenberg is highly susceptible to getting "stuck" in local minima as opposed to finding the global minima. In order for successful inverse simulation to find the global minima with this method the initial values must be varied. If the same parameter sets are found after this procedure then it can be assumed with reasonable confidence that the global minima has been identified. All subsequent findings require that you found appropriate soil hydraulic properties. I am not yet convinced that you did, I would need to see more results related to the performance of the inverse simulation.
Furthermore, as shown by the parameter values in Table 1 many of the parameters are extraordinarily tightly constrained (alpha, n, and Ks in particular). Even covers of the same (ostensible) material could easily vary by multiple orders of magnitude, yet only the lean oil sands are allowed to vary more than roughly 1 order of magnitude. The van Genuchten n parameter for the subsoil has remarkably little freedom to vary.
Also I could not seem to find any mention of root water uptake parameters, or the parameters associated with partitioning E and T from PET, or the interception constant.

*Response: The Mualem tortuosity parameter was set to 0.5 and was not optimized as the goal was to only optimize a limited set of key parameters. This is denoted by l in Hydrus-1D and defined as the pore-connectivity parameter in the hydraulic conductivity function as estimated by Mualem (1976) to be approximately 0.5 as an average for many soils. (P12L5-8)*

*We briefly discussed about the constraints of parameter ranges used in the inverse modelling in this study and the ranges shown in Table 1 in our previous response. However, we will include more information on this issue again where it seems appropriate. (P13L3-23)*

*The root water uptake parameters were obtained from previous studies on the oil sands mine reclamation covers by Huang et al. (2011a, 2015, 2017). The Feddes model parameters were set as P0 = 0 kPa; P2H = -5000 kPa; P2L = -8000 kPa; P3 = -19000 kPa; r2H = 0.5 cm/day; and r2L =0.1 cm/day for all models as obtained from the preliminary study on the same sites by Huang et al. (2017). (P12L19-22)*

P9L19 In the Discussion, I would like to have your assessment of the implications of this choice of root distribution, since it is idealized and probably incorrect.

*Response: In P8L19, we mentioned that we used the root water uptake model by Feddes et al. (1974), where the root distribution was approximated using exponential equations showing the relationship between relative root density and depth for the treatment covers since exponential root distribution was found to perform better in the near surface horizons (Li et al. 1998). However, the root distribution is affected by site conditions (Strong and La Roi 1983). We included more discussion on the implications of the choice for root distributions on the parameter optimization. (P28L4-13)*

P10L2 Unit gradient meaning free drainage?

*Response: Free drainage is not a well-defined term. If the reviewer is referring to a gravity gradient (i.e. 'unit gradient' boundary) then the answer is yes. (P14L16)*

P10L25. Are these (optimized soil parameters) the V-G parameters?

*Response: These 155 parameter sets include both VG parameters ($\theta r$, $\theta s$, $\alpha$, $n$) and saturated hydraulic conductivity (Ks). (P15L18-19)*

P11L11 I'm not clear on what you mean by discrete parameter distributions not being representative of the range of distributions

*Response: We mean the parameter distributions using discrete (not randomly selected rather fixed) percentiles (i.e. 10th, 25th, 50th, 75th, and 90th percentiles) are not able to represent the complete range of parameter distributions. (P16L7-8)*

P12L2 Just to be clear, this is the order of the soil profile? Peat/LOS/Subsoil?

*Response: In fact, the order of the soil profile is Peat/Subsoil/LOS. (P17L7-8)*

P12L6 There is evidence to suggest that some early covers generate a lot of snowmelt runoff, rather than infiltrating into the soil (e.g. Ketcheson et al., 2016), although others that suggest mostly infiltration (Nichols et al., 2016). Are there measurements or observations at this site that support your assumption. You should probably include a statement to this effect.

*Response: Thank you for this good suggestion. While runoff from the watershed would largely depend on the slope of the watershed, the amount of runoff would vary between the reclamation cover systems. Huang et al. (2015) showed an average runoff of 34 mm each year from a sloping cover (~5H:1V), while*

*other reclamation covers were flat-lying and assumed to have negligible runoff in previous studies (Alam et al. 2018; Huang et al. 2015, 2017). So, the runoff from the flat-lying reclamation cover was not simulated in this study rather incorporated in the NP rates. Therefore, the simulated NP rates represent the total water yield from the covers that may eventually reach the downgradient surface water bodies. Besides, there was no measurement to confirm which one between runoff and infiltration dominates in the reclamation cover sites. (P17L11-17)*

P12 L7 Does this method consider sublimation?

*Response: The method uses a constant that accounts for all the factors affecting the snow melt amount and varies with time. The method did not consider sublimation as intercepted snow results in the highest rates of sublimation; however, interception of snow is quite low in case of a deciduous tree (e.g. aspen). (P17L19-22)*

P13 L9 The performance of your model should not be hidden in an appendix, and it is not acceptable to just show the r2, if only a single model metric were chosen, show the Kling-Gupta Efficiency. Otherwise, the RMSE, should be shown as well as perhaps graphs over time of the simulated and observed water contents.

*Response: Please refer to Table 1 and Figure 1 of this document.*

P13L13 This seems poor justification for grouping the parameter sets together.

*Response: According to Syncrude Canada Ltd., in the final cover design the top layer might be either peat/LFH or combination of the two. The distributions of parameters for these two materials together seem reasonable to be used in the illustrative covers for long-term simulation of water balance components. Moreover, the primary purpose of this study was not to differentiate the performances of two alternate coversoils built on the two organic-rich materials. Therefore, the PLHS method was used to randomly sample from the distributions of the two materials grouped together and the distributions of parameters for these two materials together are used in the illustrative covers for long-term simulation of water balance components. (P21L11-17)*

P14L1 Why would you be interested in including temporal variability into these parameter sets? By then applying these parameter sets to long-term simulations you would be artificially accentuating the variability in net percolation and ETa.

*Response: The material properties of the treatment covers evolve with time as they vary in space. It seems important to see how these material properties would vary in time, if any, in addition to the spatial variability. (P22L5-7)*

P14L5 I don't think that you have shown enough supporting evidence to make that assertion. Did you perform this same procedure for alpha, n, theta r, and theta s?

*Response: We did this for Ks only as it was the most influential parameter for the treatment covers. We included the evidence to support our claim (P22L9-11)*

P17L2 Does this approach assume that there is no correlation between parameters?

*Response: Despite the correlation between these parameters in the form of a water retention curve (WRC), the PLHS method randomly selected these parameters without considering the correlation between them. However, the PLHS method was able to maintain those correlations when plotted as WRCs as shown in Fig. 6 and turns out to be a reliable method that captures the physical relationship between the VG parameters. (P26L9-13)*

P21L8 Are you surprised that the LAI max occupies such a narrow range for such different soil covers? Why is the range so small? Please add a comment on this, as 4.12 to 4.50 are basically the same, as far as model precision goes. Doesn't this suggests that you can get a reasonable forest growing on basically any cover? The differences in Figure 10 are so slight it seems like the same graph copied 5 times.

*Response: We added a line saying "Huang et al. (2015) showed that the increases in AET are not necessarily proportional to the incremental increases in cover thickness, rather little increment is noticed in the median AET over a climate cycle once a threshold cover thickness is passed. Therefore, it is not a surprise to observe the narrow range of LAI_max values as shown in Fig. 10." (P32L1-4)*

P22L9 and Fig11 Are there statistically significant differences?

*Response: In fact, the difference between A50 and A150 covers seems statistically significant (as boxes do not overlap), while differences between others seem statistically non-significant (as boxes overlap). Figure 11 shows why the differences between LAI values for five covers shown in Fig. 10 are not apparent. So, we can re-write the sentence as "These distributions of the LAI_max values demonstrate that the parameter variability results in slightly higher LAI_max and slightly lower uncertainty as cover thickness increases." (P32L12-14 & P33L1-2)*

P27L10 I'm not really sure I understand the value of this whole objective and section. I would remove this from the manuscript.

*Response: We removed this part in the revised manuscript as suggested while we may still use part of it (for brevity) in explaining the benefits of using the PLHS method somewhere in the manuscript. ( P41L19-21)*

P29L9 This seems to be a foregone conclusion since you have already admitted to lumping to very different materials into this category.

*Response: We admit that peat was combined with LFH, which might be the reason for the highest variability in WRC among the materials for three layers of the treatment covers. (P41L23-24)*

P29L19-20 "the results of this study help to highlight a wide range of cover performance risks that can occur when parameter variability is combined with climate, LAI, and cover thickness variability", this is really the first and only mention that the reader gets of cover performance risks. I am interested to know more, but it needs to come before that statement!

*Response: We thank RC1 for bringing this discussion issue. We talked about this in the revised manuscript as suggested. We mentioned "The parameter variability combined with climate variabilities due to GCMs and/or RCPs would cause more increased uncertainty in the future period than it appears to cause during the historical period, and it requires further investigation. The elevated water balance components as well as increased uncertainty in the simulated AET and NP rates due to combined impacts of climate and*

*parameter variability would pose increased risks to the management of water migrating through reclaimed mine waste. The risks of increased chemical loading to the downgradient waterbodies due to increased NP rates will require to be investigated under changing climate conditions." (P37L5-14)*

P29L25 AET really didn't change that much in any scenario. The data seems to conflict with the interpretation.

*Response: Table 3 demonstrates this statement clearly for the annual AET and NP for the five covers, while decreases in their ranges are shown in Figs. 13 and 14. (P37 Table 4)*

Fig. 3 I could be mistaken but it seems that the hydraulic conductivities are outside the maximum and minimum values that are seen in Table 1. The fact that some parameters are logged and others not makes it difficult to compare. The alpha values in the figure are in different units than what is in Table 1 (not the only instance of alpha values being in inconsistent units). The saturated water content of the subsoil seems to in some cases be very low. What is the cause of the inverse scheme identifying a 0.1 or 0.2 water content? That is almost certainly incorrect.

*Response: While Table 1 shows initial values and ranges of parameter search in inverse modelling for Treatment cover #10 (it is shown as an example for brevity of the manuscript), Fig. 3 shows the ranges of parameter search in inverse modelling for all 13 treatment covers (replicated in triplicate and monitored in four consecutive years). So, Table 1 is part of Fig. 3 but not showing the complete ranges for all covers. We will certainly address the inconsistency in the units of parameters and log-scale issues in the revised manuscript. In Table 1 and Fig. 3, we used the consistent units for the parameters shown in Table 1 and Fig. 3. (P13-14 Table 1 and P22 Fig.4)*
*The optimized saturated water content (theta_s) depends on the measured theta_s in the subsoil layer of the treatment covers. The measured values of maximum water content in the subsoil layer show as low as 0.1 $cm^3/cm^3$ in some years and some treatment covers.*

Fig. A2 While it seems reasonable that the subsoil and lean oil sands are grouped together. It does not seem appropriate that the peat and LFH soils are grouped. Their parameters (particularly alpha and n) differ substantially.

*Response: In the revised version, we showed the parameters for peat, LFH, subsoil, and LOS separately in Fig. A1 (as shown in Fig. 2 below). Please see our previous response on this issue for more details.*

[Figure]

*Figure 2: Frequency distributions of the optimized parameters for LFH (top panel), peat (second panel) subsoil (third panel), and LOS (bottom panel) materials (P45 Fig. A1)*

Formatting Comments
Fig. 1 Font in legend is unclear - small and resolution too poor. Figure should be improved for readability

*Response: We will definitely consider reproduction of Fig. 1 with more readability and clarity in the revised version. (P7 Fig. 1(b))*

Fig. 2 Should be improved for readability, very difficult to understand

*Response: We revised Fig. 2 with more readability and clarity for the revised manuscript.*

[Figure]

Figure 3: *Particle size distribution for (a) LFH, (b) peat, (c) subsoil, and (d) LOS materials for the treatment covers (OKC, 2009). The lines in the subplots show PSDs for different samples collected from the LFH, peat, subsoil, and LOS layers, respectively.* (P11 Fig. 2)

Fig. 9 Could probably be moved to an appendix

*Response: Sure, we moved it to the appendix in the revised manuscript. (P46 Fig. A2)*

Fig 11 could be replaced with another line for LAI+/- SD in Table 3.

*Response: Sure, we incorporated this in the revised manuscript (Table 2 in this document). (P37 Table 4)*

*Table 2: Summary of water balance components for the five illustrative covers obtained using 700 sampled parameter sets with the corresponding LAI_max values (the mean LAI_max value is shown with the corresponding standard deviation)*

| Illustrative cover | Precipitation (mm) | AT (mm) | AE (mm) | AET (mm) | NP (mm) | DS (mm) | LAI_max ± SD [-] |
|---|---|---|---|---|---|---|---|
| A50 | 426 | 297 | 99.1 | 397 | 33.3 | -3.41 | 4.12 ± 0.33 |
| A75 | 426 | 303 | 98.6 | 402 | 28.0 | -3.47 | 4.25 ± 0.30 |
| A100 | 426 | 305 | 99.8 | 405 | 25.0 | -3.53 | 4.27 ± 0.29 |
| A125 | 426 | 310 | 101 | 411 | 20.0 | -4.69 | 4.37 ± 0.27 |
| A150 | 426 | 316 | 101 | 416 | 15.0 | -5.46 | 4.50 ± 0.23 |

Fig 12&13 could be put into a single Table (there are a lot of Figures).

*Response: This is a good suggestion, but figures might be clearer to the readers than a table. We looked at the final number of figures in the revised manuscript, and we believe there is still room for these figures.*

Fig 15 could be summarized in one or two sentences in the text.

*Response: We included the summary in the revised manuscript (P41 L19-21 and P51 Fig. A6).*

*Responses to the Referee #2 (Claire Cote)*

*We appreciate the Referee #2 for her comments on our paper. Below we attach our brief response to the issues raised by the reviewer in expectation of a successful discussion.*

General Comments The objectives of this manuscript are not clear, are focused on minor modelling considerations and do not contribute valuable findings to the field of waste cover design. This is unfortunate as it forms part of a broader program of work that would bring interesting findings worth publishing. For mine reclamation covers, key objectives of the design should be that (1) it will lead to successful ecosystem restoration ( eg. plants are growing successfully) (2) the quality of the seepage does not lead to impacts on the receiving environment and (3) it addresses any other objective identified as part of design and mine closure planning. The manuscript does not clearly state how it contributes to the understanding of how the selected cover designs will meet these objectives.

*Response: The authors agree regarding the design objectives for mine reclamation covers. However; we humbly disagree with the reviewer's perspective on our paper. The purpose of the study was to highlight how climate and parameter uncertainty might be incorporated into these designs so that the variability and uncertainty associated with the reviewer's objective 1 (e.g. AET) and objective 2 (e.g. amount of NP) can be more fully characterized. In our humble opinion it appears that the reviewer has misinterpreted the primary objectives and contribution of the paper. (Please see P4 L15-19)*

Among other things, models are used to provide a simplified mathematical representation of reality so that various scenarios can be analysed. If we change the parameters in the mathematical representation, the results from the model will be different. It does not meant the reality itself has changed, so I struggle to understand what the study is trying to prove. It seems like a circular proposition. If we modify the van Genuchten parameters then of course the predicted water balance will be different. The issue is not the value of the parameters but our ability to understand how changes to the water balance can prevent the design from meeting the objectives stated above.

*Response: We agree with the reviewer's view of models; however, it would be of great value to a designer to be able to assign 'uncertainty' to a particular design (parameters or climate) since these uncertainties are directly tied to risk (i.e. cost \* expectation). You can change parameters all you like but it doesn't tell you anything about the source of the uncertainty or quantify the magnitude of this uncertainty. Again – the reviewer's comment, although correct in and of itself, makes us wonder if the reviewer understood the purpose of the paper.*

For instance, a design is developed specifically to minimize seepage but a modification in the surrounding environment leads to increased seepage and risk to the receiving environment. Statistical analysis can assist with quantifying the probability of that scenario and this is what we should be focusing on. If the scenario is likely then there is a requirement to develop mitigation options, such as installation of additional monitoring equipment to capture changes and mitigate them. The values of the hydraulic parameters in the mathematical representation are not that relevant to the mitigation of the issue.

*Response: We fully agree that the key is assigning probability to the predictions. We humbly disagree that there then is no value in assessing the potential impact that uncertainty in hydraulic parameters (and climate) have in this assessment of uncertainty.*

With advances in mathematical software, particularly Mathematica, MatLab or R, there is less and less of a need to develop specific mathematical approaches for solving soil science or environmental science

problems. With the Wolfram Language now in the public domain, it is even less justifiable. I have never heard of the Latin Hypercube Sampling, but looking at the description in the manuscript, all I would need to do in Mathematica is to Map the Range function unto the parameters. It seems very straight forward and certainly does not justify a publication. Hydrus 1D could probably be re-coded in Mathematica very quickly and the predictions would be much faster and much more stable. It does not seem that there has been much evolution in the field of vadose zone hydrological modelling in the last 20 years.

*Response: The unsupported and unjustifiable opinion of the reviewer seems to highlight either a lack of understanding or an anchored bias. If 'all I would need to do' was true then why has this type of study on sources of uncertainty in soil cover water balance been published previously? We strongly disagree with the statement that there is 'less of a need to develop specific mathematical approaches'. Mathematics and numerical models are simply tools. But we are trying to utilize these tools in a unique way to gain some insight into a critical problem in the evaluation of long-term soil cover performance.*

There would be value in re-assessing these models in light of advances in mathematical software. In terms of developing a more robust approach to test the performance of cover design, I would suggest leaching experiments on large columns, which can reproduce leaching results much more quickly than what can be obtained in the field. It is a more reliable method to investigate the impact of variations in the water balance. The greatest long term risk to the viability of soil covers and associated ecosystems is climate change with associated variations in temperature and precipitation. Column tests can provide an indication of expected changes in soil water behaviour under extreme precipitation or temperature regimes. This would be a much more robust contribution than mathematical manipulations of soil parameters. Another key gap seems to be the assessment of snow pack height on cover behaviour. This could also be investigated with column testing. Assessment against HESS criteria

*Response: Modelling, including the types of evaluation of uncertainty we have proposed, doesn't preclude but rather builds on strong field or even column based data gathering.*

*However, the reviewer's confidence in solving this problem through the use of column testing is misplaced and misinformed. Column tests and small/short-term field cover tests have proven repeatedly to be difficult to interpret relative to long-term cover performance over large spatial and temporal scales. The suggestion that snow pack height is best investigated in column tests is particularly incongruent.*

The paper is well-written, well-referenced and well-structured. Figures, tables and cross-references are fine. The key issue relates to the value of the contribution to the hydrological community. The objectives and research questions are ill defined. The paper does not present novel concepts, ideas or tools. Clear conclusion are provided but they do not contribute much to the assessment of the performance of cover designs. The description of experiments and calculations are complete and precise. The authors give proper credit to related work. The work described in the paper is part of a broader project that is very interesting. The title clearly reflects the contents of the paper. The abstract provides a concise and complete summary.The language is fluent and precise. The number and quality of references appropriate There is no need to provide details about the Latin Hypercube Sampling methodology, it is a routine calculation.

*Response: The authors want to thank the reviewer for taking the time to read the paper and provide her perspectives and comments.*

*Where there has been constructive and useful recommendations we will make appropriate changes. However, overall the comments of the reviewer appear to betray both a misunderstanding of the actual goal (and value) of the paper and/or betray some anchoring bias towards completely different approaches to study soil cover performance (e.g. not using physics based models but rather column testing).*

*We have attempted to respond positively to the apparent confusion of the reviewer regarding our contribution by further refining the following comments. We sincerely thank RC2 Claire Cote for bringing the following issues to our attention which will definitely improve the quality of our manuscript.*

**Novelty and Contribution to the hydrological community:**

*Long-term cover performance is commonly evaluated without quantification of the uncertainties which may originate from various sources including: climate, soil hydraulic parameters, vegetation index (i.e. leaf area index) etc. The use of a single set of model parameters in the design of reclamation covers precludes our ability to quantify the potential impact of uncertainties in parameters or climate and consequently makes it impossible to quantify the associated risks in performance. While our previous study (Alam et al. 2018a) investigated the impacts of climate in changing climatic conditions on the long-term cover performance, this study investigates the sources of uncertainty associated with the evaluation of long-term cover performances. This study considers a unique way to characterize the spatial and temporal uncertainty in the total parameter uncertainty utilizing field monitoring data from 13 treatment covers (replicated in triplicate and monitored in four consecutive years). The field monitoring data includes water content, soil temperature, and soil suction values recorded at various depths at each of the treatment covers. There are few instances in the literature where such a large data record is available to quantify uncertainty. It has not been attempted previously in the context of oil sands reclamation covers. While the use of Hydrus-1D inverse modelling tool to optimize soil hydraulic parameters (both VG and saturated hydraulic conductivity) is not new, the use of this tool to develop probability distributions for optimized parameter sets from multiple sites and years is novel, particularly when one of the parameters (Ks) can be directly compared to field measured distributions. Since, the risks associated with a cover design would be based on the probability distributions of water balance components, it seems reasonable to use a computationally efficient sampling method (PLHS in this case) to obtain all possible probability distributions of the optimized parameters, which was motivation for the current study. (P40L7-11 to P41L1-12)*

**Objective and research questions:**

*The key research question of this study is as follows: What is the influence of soil hydraulic parameter uncertainty on the long-term cover performance of the reclamation covers in northern Alberta, Canada. This question led us to the following study objectives: (i)Identify the most-efficient way to characterize distributions of the optimized hydraulic parameters from a physically-based water balance model for an oil sands reclamation covers in northern Alberta, Canada and (ii) Quantify relative uncertainty from various sources associated with the long-term water balance of the reclamation covers. (P4L15-19)*

**Conclusion on the assessment of cover designs:**

*We included this in our responses to RC1; however, we are repeating it here for ease of reference. We added to our conclusion, "
[revised manuscript text omitted]